# Characterization of sequence determinants of enhancer function using natural genetic variation

**Marty G Yang[1,2†], Emi Ling[1†‡], Christopher J Cowley[1§], Michael E Greenberg[1*], Thomas Vierbuchen[3,4*]**

[1]Department of Neurobiology, Harvard Medical School, Boston, United States; [2]Program in Neuroscience, Harvard Medical School, Boston, United States; [3]Developmental Biology Program, Sloan Kettering Institute for Cancer Research, New York, United States; [4]Center for Stem Cell Biology, Sloan Kettering Institute for Cancer Research, New York, United States

**\*For correspondence:**
meg@hms.harvard.edu (MEG);
vierbuct@mskcc.org (TV)

†These authors contributed equally to this work

**Present address:** ‡Department of Genetics, Harvard Medical School, Boston, United States; §Laboratory of Mammalian Cell Biology and Development, Howard Hughes Medical Institute, The Rockefeller University, New York, United States

**Competing interest:** The authors declare that no competing interests exist.

**Abstract** Sequence variation in enhancers that control cell-type-specific gene transcription contributes significantly to phenotypic variation within human populations. However, it remains difficult to predict precisely the effect of any given sequence variant on enhancer function due to the complexity of DNA sequence motifs that determine transcription factor (TF) binding to enhancers in their native genomic context. Using $F_1$-hybrid cells derived from crosses between distantly related inbred strains of mice, we identified thousands of enhancers with allele-specific TF binding and/ or activity. We find that genetic variants located within the central region of enhancers are most likely to alter TF binding and enhancer activity. We observe that the AP-1 family of TFs (Fos/Jun) are frequently required for binding of TEAD TFs and for enhancer function. However, many sequence variants outside of core motifs for AP-1 and TEAD also impact enhancer function, including sequences flanking core TF motifs and AP-1 half sites. Taken together, these data represent one of the most comprehensive assessments of allele-specific TF binding and enhancer function to date and reveal how sequence changes at enhancers alter their function across evolutionary timescales.

## Editor's evaluation

Here, the authors used multiple F1 crosses and the resulting embryonic fibroblasts to perform molecular profiling with ATAC-seq and a combination of ChIP-seq, Hi-ChIP, and CUT&RUN on multiple modified histones and transcription factors proteins. These important results are a convincing resource for quantifying allelic bias in protein-DNA binding and chromatin accessibility.

## Introduction

Genome sequencing efforts have uncovered large numbers of sequence variants associated with phenotypic variation in complex traits in human populations. A significant proportion of these genetic variants occur within the ~2–3 x $10^6$ *cis*-regulatory elements (CREs) predicted across the human genome (*Carroll, 2008*; *Maurano et al., 2012*; *Pickrell, 2014*; *Li et al., 2016*; *Boyle et al., 2017*). The majority of these CREs are thought to be gene-distal enhancers that potentiate gene transcription in a cell type- or cell state-specific manner (*Keilwagen et al., 2019*). However, pinpointing the specific sequence changes in CREs that impact expression of linked genes, and downstream molecular and cellular phenotypes, remains a critical challenge in the field (*Farh et al., 2015*; *Nasser et al., 2021*; *Lappalainen and MacArthur, 2021*). More specifically, it is difficult to reliably distinguish functional

**eLife digest** There are hundreds of different types of cells in the body. Each one performs a unique role, but they all share the same genes. Sequences of the genetic code called enhancers decide which genes each cell uses. Enhancers work like genetic switches: to turn a gene on, proteins called transcription factors assemble on an enhancer. Each transcription factor recognises a short sequence on the enhancer, and several distinct transcription factors work together to promote the activatation of a gene.

The relationship between transcription factors, enhancers, and gene activation is complex. The specific genetic sequences of enhancers differ between species, changing the way these genetic switches work. But scientists are not yet able to reliably predict the effects of small changes in the DNA sequence of an enhancer. One way to tackle this problem is to look at different versions of the same enhancers side by side to see how small mutations change their behaviour.

Mammalian cells generally carry two copies of each chromosome (the molecules that contain the genetic code), one inherited from each parent. Each of the two copies carries the same genes and enhancers, but there are many small differences in the DNA sequences of enhancers between the chromosomes inherited from each parent, which can potentially alter their function

Yang, Ling et al. generated cells from mice that come from different inbred strains, which are similar to purebred dogs. By breeding two distinct inbred mouse strains together that are very different from one another, they generated a panel of hybrid mouse cell lines that have a relatively large number of differences in their DNA sequence between the maternal and paternal chromosomes.

Looking at the different versions of each enhancer side-by-side revealed thousands of single letter changes in the DNA sequence of enhancers that changed how they work. Mutations affecting the binding site of one transcription factor within an enhancer can indirectly affect the binding of other types of transcription factors. Yang, Ling et al. found that if a transcription factor could no longer find its place on an enhancer, it stopped others from binding even if their own places had not changed. Sometimes, mutations on either side of the binding sequences also affected transcription factor binding. This suggests a more complex relationship than previously thought may exist between the DNA sequence of an enhancer and the transcription factors that bind to it.

Spotting the differences caused by mutations could help further the efforts of scientists to read and write the genetic code. This could have many benefits. It would allow scientists to control natural or artificial genes, and to predict the effects of genetic changes that are identified in humans with genetic diseases. This might improve genetic experiments, medical screening, gene therapy, and our understanding of evolution.

---

sequence variants at CREs among a large excess of neutral variants. As a result, functional assays, such as plasmid-based reporters, have typically been used to assess the impact of individual sequence variants within enhancers. Since these experiments can be laborious to perform and subject to experimental artifacts, a better method for defining sequence-to-function relationships for enhancers in their endogenous genomic context could have a transformative effect on our ability to identify functional sequence variants in CREs in human genomes (*Klein et al., 2020*; *Levo and Segal, 2014*).

Enhancers are typically bound by ~4–5 TFs that recognize short sequence motifs (~6–12 nucleotides; *Bilu and Barkai, 2005*; *Meuleman et al., 2020*). TFs function as adaptor proteins to recruit transcriptional regulatory complexes to enhancers, leading to potentiation of transcription at associated gene promoters. Enhancer activity is highly cell type-specific, and this specificity of function is encoded by the type and arrangement (also known as regulatory grammar) of TF-binding motifs within each enhancer (*Zeitlinger, 2020*; *Jindal and Farley, 2021*; *Long et al., 2016*). Enhancers that control transcription in specific cell types are often bound by combinations of TFs that occur uniquely in that cellular context (*Spitz and Furlong, 2012*; *Wei et al., 2018*). This complicates efforts to identify generalizable features that can be used to prioritize enhancer sequence variants *in silico* (*Kasowski et al., 2010*; *Ding et al., 2014*; *Tehranchi et al., 2016*).

Although enhancers cannot be defined by a singular set of sequence features, they do exhibit stereotyped chromatin features that can be measured genome-wide, such as chromatin accessibility (controlled by TF and co-factor binding), histone post-translational modifications (e.g. H3K4me1/2

and H3K27ac), and bi-directional transcription of short enhancer RNAs (*Heintzman et al., 2007*; *Boyle et al., 2008*; *Creyghton et al., 2010*; *Kim et al., 2010*; *Rada-Iglesias et al., 2011*). These chromatin signatures have been used extensively to identify millions of putative enhancers in a wide range of cell types and across different stages of organismal development (*Kundaje et al., 2015*). While mapping genomic regions that function as enhancers has facilitated the identification of DNA-binding motifs enriched at enhancers in different cell types, these data have not proven to be sufficient to generate quantitative, predictive models for TF binding and enhancer function from available databases of enhancer sequences (*Deplancke et al., 2016*).

Sequence variants that disrupt TF binding can be highly informative for identifying sequences critical for the control of enhancer function in specific cell types (*Wittkopp and Kalay, 2011*; *Albert and Kruglyak, 2015*; *Lappalainen, 2015*; *Pai et al., 2015*; *Vierbuchen et al., 2017*). Our lab and others have previously used the extensive genetic variation present among inbred mouse strains to conduct 'mutagenesis screens' of enhancer sequences in their native chromatin context (*Heinz et al., 2013*; *Vierbuchen et al., 2017*; *Wong et al., 2017*; *van der Veeken et al., 2019*). By crossing highly divergent inbred mouse strains to generate $F_1$ hybrids, it is possible to directly compare the activity of two alleles of each enhancer locus within the same cellular environment.

Using mouse embryonic fibroblasts (MEFs) derived from two distinct inbred strains, we found that the binding of AP-1 TFs is required for chromatin accessibility and activity at many active enhancers in fibroblasts (*Vierbuchen et al., 2017*). However, we also observed that many instances of allele-specific AP-1 binding cannot be readily explained by sequence variants within AP-1 motif(s). These data indicated that, at many enhancer loci, sequence features outside of AP-1 TF-binding sites contribute to AP-1 binding. In this previous study, we observed an enrichment of variants in motifs for putative collaborating TFs (e.g. TEAD) at these sites, but the nature of this collaborative relationship with AP-1 remained to be defined. Both AP-1 and TEAD are broadly expressed and have critical roles in mediating signal-dependent transcription downstream of the Ras/MAPK and Hippo/YAP/TAZ pathways, respectively. Consistent with our findings, the co-occurrence of AP-1 and TEAD motifs at enhancers has also been noted in a variety of human tumor cells, providing support for the idea that AP-1 and TEAD coordinately regulate cell fate and proliferation. Therefore, delineation of the sequence features that determine the binding of AP-1 and TEAD TFs to enhancers could provide insight into enhancer function across a broad range of cellular contexts (*Zanconato et al., 2015*).

In the present study, we carried out an extensive allele-specific analysis of chromatin state (ATAC-seq, H3K27ac, H3K4me1/2, and H3K4me3) and TF binding (Fos, Tead1, and CTCF) in $F_1$-hybrid MEFs derived from crosses between C57BL/6 J mice and a panel of nine inbred mouse lines, including several wild-derived inbred strains from distinct sub-species of mice that contain a high frequency of SNPs/indels (1 in every ~85–170 bp) compared to C57BL/6 J mice. Using these genetically divergent strains, we examined the frequency and distribution of SNPs/indels at thousands of enhancers with allele-specific chromatin features and/or TF binding patterns. We found that sequence variants within the central ~50 bp of enhancer sequences were most likely to lead to an allele-specific change in enhancer activity. These data also revealed that AP-1 binding is often required for TEAD TF binding to enhancers, whereas TEAD is generally dispensable for AP-1 TF binding. This result is consistent with a model in which AP-1 TFs function as pioneer factors and facilitate binding of additional TFs during enhancer selection in fibroblasts. An analysis of our allele-specific data revealed that sequence features, such as partial AP-1 motifs and nucleotide sequences flanking core AP-1 binding motifs, also contribute to enhancer function. These findings provide new insight into how AP-1 TFs function at enhancers in fibroblasts, and suggest that across other cell types, AP-1 TFs may employ similar collaborative binding mechanisms at enhancers. In addition, our data provide new insight into the crosstalk between Ras/MAPK and Hippo/YAP/TAZ/TEAD signal-dependent gene expression and suggest that Ras/MAPK-induced AP-1 can play an instructive role in determining the output of Hippo/YAP/TAZ/TEAD-dependent transcriptional programs.

## Results
### Mapping TF binding and CREs in $F_1$-hybrid MEFs
To identify genetic variants that modulate TF binding and/or chromatin state at CREs, we isolated MEFs from male $F_1$-hybrid embryos derived from crosses between C57BL/6 J females and males from

nine distinct inbred mouse strains, including four wild-derived inbred strains (CAST/EiJ, MOLF/EiJ, PWK/PhJ, and SPRET/EiJ) that have a high frequency of SNPs/indels compared to C57BL/6 J mice (*Figure 1A*; *Supplementary file 1*). Genome sequencing data is available for each inbred strain, meaning that we can query up to ten distinct alleles at each CRE sequence for differences in TF binding and/or *cis*-regulatory activity (*Keane et al., 2011*). To identify potential differences in CRE function that result from sequence variants between maternal (C57BL/6 J) and paternal chromosomes, we generated the following allele-specific datasets from the four wild-derived inbred $F_1$-hybrid strains: chromatin features associated with *cis*-regulatory function (ATAC-seq, H3K4me1/2, H3K4me3, and H3K27ac), occupancy of TFs that bind many CREs in fibroblasts (Fos and Tead1), putative insulator elements (CTCF), and gene transcription levels (chromatin-associated RNA-seq; *Figure 1B–F*, *Figure 1—figure supplement 1A*; *Supplementary file 2*). For the remaining $F_1$-hybrid lines (129S1/SvImJ, A/J, BALB/cJ, DBA/2 J, and NOD/ShiLtJ), we only performed CUT&RUN for Fos and H3K27ac (*Supplementary file 2*). H3K27ac Hi-ChIP was also performed in C57BL/6 J MEFs to link active enhancers to putative target genes and to other active CREs (*Supplementary file 3*). All experiments were conducted under two distinct conditions: (1) MEFs that were growth arrested in $G_0$ by serum starvation and (2) serum-starved MEFs that were re-stimulated with serum for 90 min. These defined conditions reduce technical variability between samples by synchronizing cells in the population at a specific stage of the cell cycle and allow us to measure the binding of TFs that are induced by serum stimulation, such as AP-1 TFs, at the peak of their activity (*Vierbuchen et al., 2017*).

From each of the two alleles in $F_1$-hybrid lines, we identified putative primed CREs (ATAC-seq summits that lack H3K27ac) and active CREs (ATAC-seq summits overlapping H3K27ac peaks) (*Figure 1—figure supplement 2A*). For all active CREs, we classified sites as either gene-proximal (promoters) or gene-distal (putative enhancers) based on their distance to the nearest annotated TSS (*Figure 1—figure supplement 2C*). In total, we found 76,517 unique genomic loci defined as active CREs from the nine $F_1$ hybrids surveyed, and 50.4% of allele pairs at these sites harbor SNP(s) within +/-60 bp of the ATAC-seq summit used to define each enhancer locus.

## Identification of allele-specific CREs in $F_1$-hybrid MEFs

In aggregate, across all nine $F_1$-hybrid lines, 24.4% of pairs of active enhancer alleles on autosomes show a statistically significant difference in H3K27ac levels between maternal (C57BL/6 J) and paternal alleles (*Figure 1C*, *Figure 1—figure supplement 2B*). Among these allele-specific sites, 56.2% and 15.3% exhibit a >2 fold and >4 fold difference in H3K27ac signal, respectively. To determine whether differences in H3K27ac between alleles are associated with changes in transcription of the gene that they regulate, we first identified high-confidence enhancer-TSS interactions using H3K27ac Hi-ChIP data, and then examined whether transcription of the linked gene was higher on the chromosome with the active enhancer allele. We found that allele-specific enhancers are more likely to interact with genes that exhibit allele-specific transcriptional differences than enhancers that have similar levels of H3K27ac on each allele (14.5% and 9.1% of active enhancers with detectable H3K27ac Hi-ChIP loops with an active promoter, respectively; *Figure 1—figure supplement 2D*). This suggests that allele-specific differences in H3K27ac are indicative of functional differences in the transcriptional regulatory activity of enhancers, consistent with findings from previous studies (*Creyghton et al., 2010*; *Arnold et al., 2013*; *Fulco et al., 2019*).

Compared to H3K27ac levels at enhancers, levels of promoter-associated histone modification H3K4me3 (3.6%) and gene-distal binding of CTCF (2.6%) are less likely to exhibit significant differences between alleles (*Figure 1D–E*). These data are consistent with previous studies suggesting that promoters and CTCF-binding sites are more likely to be functionally conserved than enhancers when compared across groups of distantly related species (*Schmidt et al., 2012*; *Villar et al., 2015*; *Fudenberg and Pollard, 2019*). Furthermore, for each class of active CREs, we found that the frequency of sites with allele-specific H3K27ac signal is proportional to the frequency of SNPs between maternal and paternal alleles (*Supplementary file 4*). We noted that the number of genes with an allele-specific skew in expression level per strain also scaled with the total number of SNPs/indels relative to C57BL/6 J in the given strain (*Figure 1—figure supplement 1B*; *Supplementary file 5*).

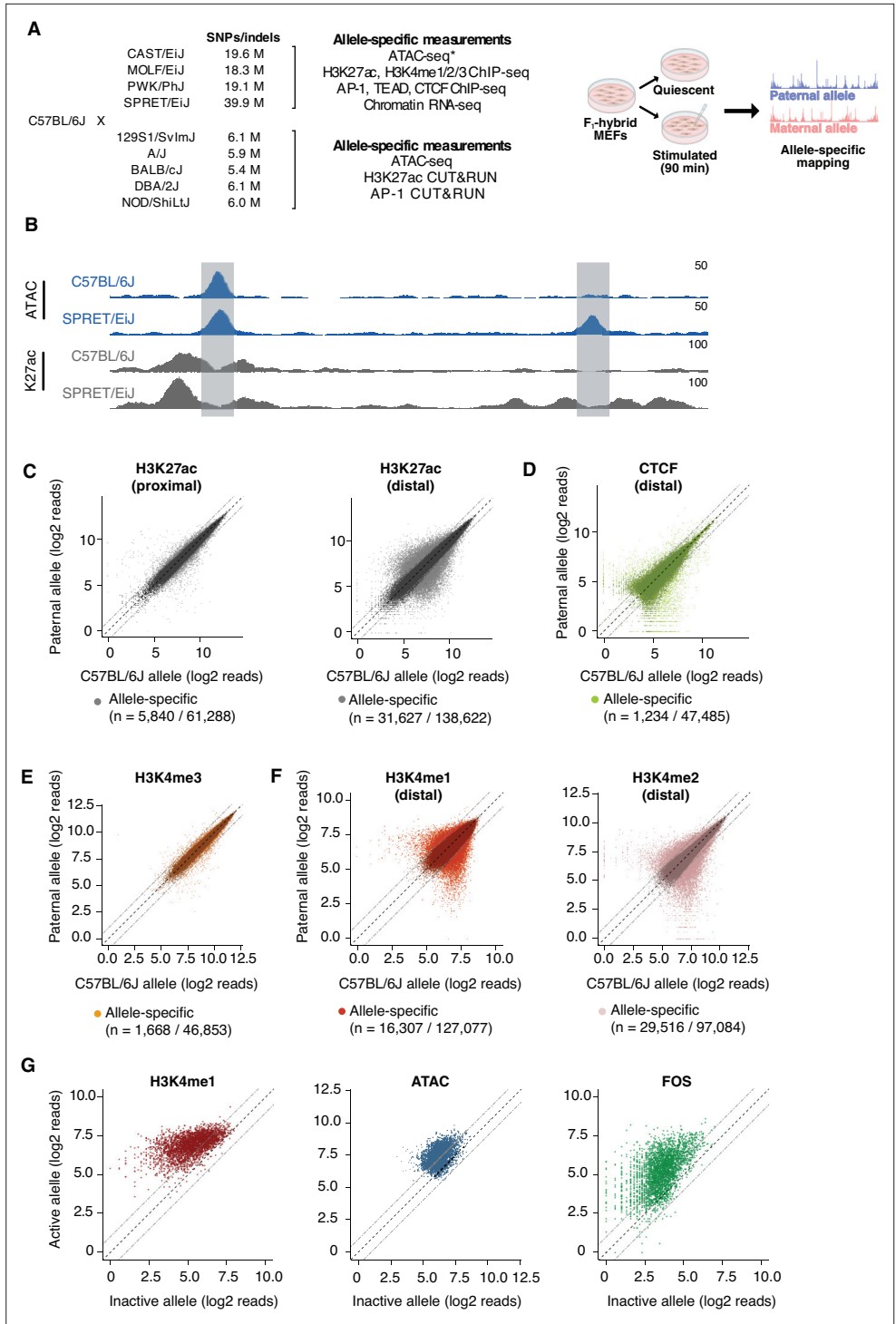

**Figure 1.** Allele-specific mapping of CREs and TF binding. (**A**) F$_1$-hybrid male MEFs were derived from crosses between female C57BL/6 J mice and male mice from a panel of inbred mouse strains. Experiments were performed in quiescent (0 min) and serum-stimulated (90 min) MEFs from at least two independent male embryos as biological replicates for each assay. Reads were mapped to either the maternal or paternal allele to quantify chromatin state and TF binding at CREs in an allele-specific manner. For wild-derived inbred strains, ATAC-seq data was generated using MEFs from corresponding parental lines and compared with chromatin accessibility in C57BL/6 J MEFs. Similarly, H3K27ac Hi-ChIP data was obtained only from starved and serum-stimulated MEFs from C57BL/6 J mice. All other genomic data indicated herein were obtained using MEFs derived from male F$_1$-hybrid embryos. (**B**) Example genome browser track of a locus (chr5:147,587,473–147,599,697 in mm10 genome) with an

*Figure 1 continued on next page*

*Figure 1 continued*

allele-specific enhancer (indicated in gray, on the right) in C57BL/6 J x SPRET/EiJ F$_1$-hybrid MEFs. Normalized read densities for ATAC-seq and H3K27ac ChIP-seq for each allele are shown. (**C–F**) Scatterplots of maternal (C57BL/6 J) and paternal allele-specific signal for histone modifications and CTCF binding (n=61,288 proximal H3K27ac, n=138,662 distal H3K27ac, n=47,485 distal CTCF, n=46,853 proximal H3K4me3, n=127,077 distal H3K4me1, and n=97,084 distal H3K4me2 allele pairs, respectively). Points indicated in light and dark colors represent peaks with and without a significant skew in signal between alleles, respectively (FDR <0.1 with DESeq2). CTCF and H3K4me3 levels were less likely to show an allele-specific skew in signal, in comparison with H3K27ac levels at active enhancers (Fisher's exact test, p<2.2 x 10$^{-16}$ for CTCF, p<2.2 x 10$^{-16}$ for H3K4me3). (**G**) Scatterplot of allele-specific H3K4me1, ATAC-seq, and Fos binding signal at top decile of allele-specific enhancers, comparing signal from the active and inactive alleles (defined based on relative H3K27ac levels) to one another (n=13,862 allele pairs).

The online version of this article includes the following figure supplement(s) for figure 1:

**Figure supplement 1.** Allele-specific quantification of chromatin-associated RNA-seq data.

**Figure supplement 2.** Properties of active CREs and regulated target genes in fibroblasts.

## Identification of sequence features that impact enhancer selection and activation

Several mechanisms have been proposed for how TFs initially bind enhancers leading to enhancer activation and the expression of genes that were previously silent. It remains unclear whether TF binding is sufficient to displace histone octamers at nucleosomal enhancers or if TF-mediated recruitment of additional co-regulatory proteins, such as chromatin remodeling complexes, is also required (*Lidor Nili et al., 2010*; *Paakinaho et al., 2017*; *Johnson et al., 2018*). For instance, it is thought that H3K4me1/2 deposition is indicative of enhancers that have been partially activated or primed (*Heintzman et al., 2007*). However, it is not known whether the majority of these primed sites only become active later in development (i.e. subsequently gain H3K27ac; *Creyghton et al., 2010*; *Rada-Iglesias et al., 2011*; *Bonn et al., 2012*; *Bogdanovic et al., 2012*), or if they typically are fully activated in a single step (i.e. concurrently gain H3K4me1/2 and H3K27ac), such as upon the binding of signal-dependent TFs or during cellular differentiation (*Kaikkonen et al., 2013*; *Ostuni et al., 2013*). To address these hypotheses, we examined our allele-specific H3K4me1 and H3K4me2 ChIP-seq datasets, which contain thousands of allele pairs that have significant differences in these histone modifications (*Figure 1F*). Our previous work suggested that disruption of AP-1 TF binding results in the loss of histone marks associated with both primed and active enhancers (*Vierbuchen et al., 2017*), but whether this feature is generally applicable for all enhancers (independent of AP-1 binding) and whether there are mutations that inactivate enhancers without affecting H3K4me1/2 levels were unresolved. To assess whether the priming and activation of enhancers are genetically separable processes, we focused on H3K4me1/2 levels at enhancers with the greatest difference in H3K27ac levels between alleles. We observed that 70.1% of enhancers in the top decile of allele-specific enhancers have a significant and >2-fold concordant loss of H3K4me1 on the inactive allele, compared to 0.6% of the bottom decile of allele-specific enhancers (peaks with the smallest, statistically significant fold changes in H3K27ac levels between alleles). Chromatin accessibility and AP-1 binding exhibit similar changes to H3K4me1 at enhancers with strongly allele-specific H3K27ac (*Figure 1G*). Together, these data reveal that few if any SNPs/indels cause a significant loss of enhancer H3K27ac and maintain strong enrichment of H3K4me1/2 and chromatin accessibility. Thus, our data is consistent with a model in which enhancer priming/selection and activation are not separable steps mediated by distinct TF-binding events at enhancers in MEFs.

## Contribution of *cis*- and *trans*-acting effects on enhancer activity

In F$_1$ hybrids, both enhancer alleles are exposed to the same nucleoplasmic environment, and thus observed differences between the two alleles are generally considered to be caused directly by local SNPs (i.e. SNPs within the ~200 bp sequence of chromatin accessibility at the CRE in question). However, each enhancer allele is also located within a *cis*-regulatory unit or topologically associated domain (TAD), which contains additional genetic changes outside of the enhancer itself that could potentially impact TF binding or chromatin state at the enhancer in an allele-specific manner (*Kilpinen et al., 2013*; *Grubert et al., 2015*). These 'locus-scale' *cis*-acting mechanisms could include: (1) sequence variants in other CREs at the same locus that interact with an enhancer in 3D, (2) gains

or losses in CTCF-binding sites that influence 3D interactions between CREs within the *cis*-regulatory unit associated with that enhancer, (3) structural variants that disrupt the organization of the locus such that the enhancer is subject to different 3D interactions, and (4) variation in repeat elements (e.g. LINEs, SINEs) within the locus that are not generally well annotated in genomic datasets (*Ou et al., 2019*). Another possible explanation for allele-specific activity of CREs is parent-of-origin specific imprinting. We excluded CREs at known imprinted loci from subsequent analyses due to the differing nature of this type of allele-specific transcriptional regulation.

To quantify the relative impact of these *cis*-acting, locus-level mechanisms, we analyzed sequencing reads from our allele-specific histone modification datasets, which typically flank the functional CRE sequence and can thus be mapped to one allele or the other even when there are no SNPs present in the accessible chromatin window at enhancers (*Figure 2A*, *Figure 2—figure supplement 1B*). We reasoned that enhancers lacking SNPs/indels should only show an allelic skew in H3K27ac levels when these aforementioned non-local mechanisms significantly contribute to the function of those enhancers. Only 9.1% of enhancers that have no SNPs/indels in their central 150 bp (centered on the ATAC-seq summit used to initially define the CRE) exhibit a significant, allele-specific, >2-fold skew in H3K27ac levels on flanking nucleosomes, compared to 22.1% of enhancers with SNPs/indels (*Figure 2F*). This result suggests that it is relatively rare for SNPs outside the enhancer sequence itself to influence the function of the enhancer in question. In addition, allele-specific 0-SNP enhancers are not situated significantly closer (than H3K27ac-matched shared 0-SNP enhancers) to an allele-specific CTCF peak (*Figure 2—figure supplement 1A and C*), suggesting that rearrangement of CTCF-dependent TAD boundaries is not a major contributor to allele-specific differences in enhancer activity at these sites. However, among 0-SNP enhancers, those with allele-specific H3K27ac signal were more likely to be located near another allele-specific enhancer that has at least one SNP/indel (median = 48,623 bp and 75,664 bp for allele-specific and shared 0-SNP enhancers, respectively; *Figure 2—figure supplement 1D*). In contrast, allele-specific and shared 0-SNP enhancers did not exhibit significant differences in their proximity to active CREs in general (*Figure 2—figure supplement 1E*). Consistent with these findings, we observed that allele-specific 0-SNP enhancers are frequently located in enhancer clusters with another allele-specific enhancer (within ~1–2 kb apart). In such cases, it is difficult to rule out the possibility that the quantification of H3K27ac-marked nucleosomes flanking the 0-SNP enhancer is not simply detecting diffuse signal from other enhancer(s) in the cluster (*Figure 2B*). Furthermore, based on H3K27ac Hi-ChIP, we rarely observed 0-SNP allele-specific enhancers connected via a long-range loop (e.g. >10 kb) with another allele-specific SNP/indel-containing enhancer in the same TAD. Thus, while previous studies have observed that allele-specific enhancers tend to be highly interconnected with other allele-specific enhancers (*Prescott et al., 2015*; *Link et al., 2018*), which has been interpreted to suggest that CREs within the same topological domain can modulate each other's function, our data indicate that locus-scale, *cis*-acting mechanisms exert limited effects on enhancer activity.

Next, we examined the extent to which *trans*-acting effects contribute to changes in *cis*-regulatory function and gene expression between each of the $F_1$-hybrid MEF lines. *Trans*-acting mechanisms should, in principle, affect each allele within the same $F_1$ hybrid equally, but genetic variation between the distinct $F_1$-hybrid MEF lines could confound quantitative comparisons of allele-specific enhancer function between each of the $F_1$ hybrids. To measure inter-$F_1$, *trans*-acting differences, we examined chromatin state and gene expression on the C57BL/6 J X-chromosome, which is present across all $F_1$-hybrid lines. Applying the same criteria that we had used for defining allele-specific CREs on autosomes, we did not observe any CREs with significantly different H3K27ac levels on the C57BL/6 J X-chromosome between $F_1$ hybrids (*Figure 2—figure supplement 2A*). A similar analysis of chromatin-associated RNA-seq data revealed that expression of a small subset of C57BL/6 J X-chromosome genes differed significantly between different $F_1$-hybrid strains. For example, 9.3% of expressed genes on the C57BL/6 J X-chromosome differed by >2-fold in expression between C57BL/6 J x CAST/EiJ and C57BL/6 J x SPRET/EiJ MEFs (*Figure 2—figure supplement 2B*). This includes a number of genes critical for transcriptional regulation across the genome, such as *Smarca1,* which is expressed at ~2-fold lower levels in C57BL/6 J x SPRET/EiJ hybrid MEFs compared to all other $F_1$-hybrid MEFs we surveyed. Taken together, these data suggest that *trans*-acting effects have a limited impact on histone modification levels at CREs and on gene transcription across these distinct $F_1$-hybrid strains. Therefore, for some subsequent analyses, we chose to merge chromatin and TF-binding data from allele pairs across different $F_1$-hybrid lines.

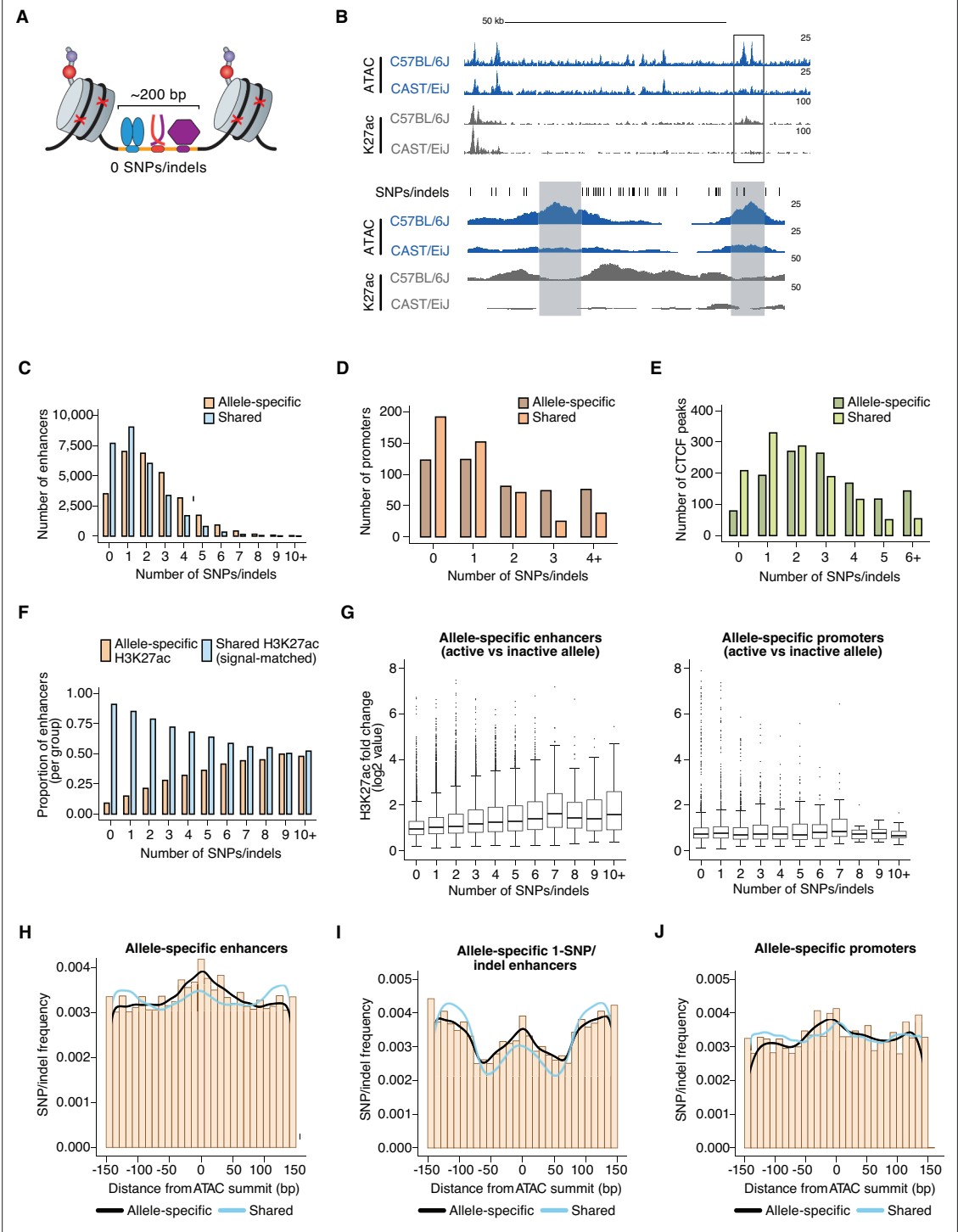

**Figure 2.** Number and position of genetic variants at allele-specific CREs. (**A**) Schematic depicting TF-bound enhancer with zero SNPs/indels (denoted by red X's) in the transposase-accessible CRE region (indicated in orange). Nucleosomes flanking both ends of the accessible region at active enhancers are marked by histone post-translational modifications, which are used as proxies for the transcriptional state of each enhancer. DNA sequences in these flanking regions tend to also be less conserved that sequences within enhancers themselves, thus often allowing sequencing reads to be correctly assigned to one of two genomes in $F_1$-hybrid cells in the absence of SNPs/indels within enhancer sequences. (**B**) Example genome browser track of a locus (chr11:113,290,106–113,416,149 (top) and chr11:113,350,775–113,356,042 (bottom) in mm10 genome) with an allele-specific 0-SNP/indel enhancer (indicated in gray, on the left) within an enhancer "cluster" in C57BL/6 J x CAST/EiJ $F_1$-hybrid MEFs. The 0-SNP enhancer is situated <2 kb from a SNP/indel-containing enhancer (indicated in green) within the same cluster. Normalized read densities for ATAC-seq and H3K27ac ChIP-seq for each allele are shown. Tick marks indicate positions of annotated SNPs/indels that distinguish the C57BL/6 J and CAST/EiJ genomes. (**C–E**) Histogram of number

*Figure 2 continued on next page*

*Figure 2 continued*

of SNPs/indels present within the central 150 bp of allele-specific and signal-matched, shared active enhancers, promoters, and putative insulators (mean = 2.36 and 1.57 SNPs/indels for enhancers, respectively, two-tailed unpaired t-test, $P<2.2 \times 10^{-16}$ for enhancers; mean = 1.84 and 1.09 SNPs/indels for promoters, respectively; two-tailed unpaired t-test, $P=7.9 \times 10^{-4}$ for promoters; mean = 2.95 and 2.03 SNPs/indels for insulators, respectively; two-tailed unpaired t-test, $p<2.2 \times 10^{-16}$ for insulators). Shared enhancers were randomly subsampled such that they were signal-matched to the active allele signal from the total set of allele-specific enhancers. (**F**) Proportion of enhancers that show allele-specific and >2 fold difference in signal, plotted as a function of the number of SNPs/indels present within their central 150 bp. Shared enhancers were randomly subsampled such that they were signal-matched to the active allele signal from the total set of allele-specific enhancers. (**G**) Box and whisker plot of H3K27ac fold changes between active and inactive alleles for allele-specific enhancers and promoters, plotted as a function of the number of SNPs/indels present within their central 150 bp. (**H–J**) Frequency of SNPs/indels at positions relative to ATAC-seq summits for allele-specific (and >2-fold) versus signal-matched, shared active enhancers, 1-SNP active enhancers, and H3K4me3-marked promoters (Pearson's Chi-squared test, $p<2.2 \times 10^{-16}$ for enhancers, $p=5.4 \times 10^{-5}$ for 1-SNP enhancers, $p=2.6 \times 10^{-9}$ for promoters). Mean number of SNPs within central 150 bp of enhancers: 4.468 for enhancers with allele-specific H3K27ac levels, 3.203 for signal-matched enhancers with shared H3K27ac levels.

The online version of this article includes the following figure supplement(s) for figure 2:

**Figure supplement 1.** Features of 0-SNP enhancers and evolutionary conservation of fibroblast CREs.

**Figure supplement 2.** Quantification of *trans*-acting effects on enhancer H3K27ac levels and chromatin-associated RNA-seq values.

## Distribution of genetic variants that influence *cis*-regulatory function

To characterize the types of variants that occur within CREs that cause changes in enhancer activity, we started by examining active enhancers with the largest differences in H3K27ac between alleles. We reasoned that these enhancers would contain large-effect, loss-of-function mutations on one allele, which could reveal TF-binding sites likely required for enhancer function. Across nine F1-hybrid strains, we identified a total of 29,185 pairs of enhancer alleles with a significant and >2-fold difference in H3K27ac levels between alleles. Allele-specific enhancers have a significantly higher frequency of SNPs/indels than H3K27ac signal-matched shared enhancers (*Figure 2C*). Moreover, enhancers with a greater number of genetic variants are more likely to show larger quantitative differences in H3K27ac levels (*Figure 2F–G*). Across many mammalian species, loci comprising allele-specific enhancers in $F_1$-hybrid MEFs exhibit slightly less evolutionary sequence conservation than those located at shared enhancers (*Figure 2—figure supplement 1F*). On the other hand, active promoters and gene-distal CTCF peaks tolerate, from a functional standpoint, a greater number of SNPs/indels than enhancers (*Figure 2D–E*).

We also hypothesized that the location of genetic variants within the enhancer is likely to impact whether a given SNP/indel affects enhancer function. To explore this further, we examined the distribution of SNPs/indels relative to the center of the accessible chromatin region at pairs of enhancer alleles with allele-specific or shared H3K27ac levels. This revealed an enrichment of SNPs/indels within a 150 bp window centered on the ATAC-seq summit at allele-specific enhancers (*Figure 2H*). In contrast, there was not a similar enrichment of SNPs/indels in allele-specific H3K4me3 peaks at promoters (*Figure 2J*). We also examined enhancer loci with a single SNP/indel present, since these sites can inform us about genetic variants that are sufficient to block enhancer function. We found that 14.6% of 1-SNP/indel enhancers show an allele-specific and >2fold skew in H3K27ac levels, and we observed a more focal enrichment directly at the central region of the enhancer (which we define as the middle ~100 bp of the <200 bp accessible chromatin window) of SNPs/indels at allele-specific enhancers (*Figure 2I*). Together, these data suggest that the central region is most likely to harbor SNPs/indels that significantly modulate chromatin state at enhancers.

## Identification of TF-binding motifs required for enhancer activity in MEFs

We next sought to determine candidate TF motifs that are required for enhancer activity in MEFs. Analysis of the top decile of active MEF enhancers (based on relative H3K27ac levels) in the C57BL/6 J genome using the KMAC algorithm (*Guo et al., 2018*) generated an output of several k-mers (i.e. nucleotide sequences of k length) that we manually matched to known binding motifs for several TF families (AP-1, TEAD, and ETS; *Figure 3—figure supplement 1A*; *Supplementary file 6*). For AP-1, the k-mer identified by KMAC (5′-VTGACTCAB-3′; V indicates A/C/G and B indicates C/G/T) includes the known core AP-1 site (known as a TRE; TGASTCA; S indicates C/G), which is bound by heterodimers of Fos and Jun family TFs or homodimers of Jun family members (*Risse et al., 1989*; *Eferl and*

*Wagner, 2003*). VTGACTCAB is the most enriched k-mer at active fibroblast enhancers (30.8%, versus 1.6% of control sequences; AUC = 0.450). AP-1 TFs bind DNA as dimers, with the basic leucine zipper (bZIP) DNA-binding domain of each Fos/Jun monomer recognizing half of a palindromic consensus motif. Because k-mer based motif representations make it possible to capture internucleotide dependencies (that are lost in position weight matrix (PWM) representations of TF-binding motifs), we were able to observe that certain flanking nucleotides on either side of the TRE are strongly depleted from bound AP-1 motifs relative to all occurrences genome-wide (i.e. T and A were depleted from the nucleotide on the 5' and 3' ends of the AP-1/TRE motif, respectively).

KMAC also identified an enriched k-mer (5'-GGAATK-3'; K indicates G/T) that matches the known core binding motif for the TEAD family of TFs (GGAAT; *Farrance et al., 1992*) and includes an additional restricted nucleotide on the 3' end (10.9%, versus 0.6% of control sequences; AUC = 0.289). TEAD TFs are broadly expressed in developing and adult cell types and function as transcriptional effectors of the Hippo/YAP/TAZ signaling pathway that regulates cell growth and proliferation (*Chen et al., 2010*).

We observed a similar enrichment of AP-1 and TEAD k-mers at both constitutively active enhancers and enhancers that control transcription of late-response genes activated by serum stimulation (identified in *Vierbuchen et al., 2017*; *Figure 3—figure supplement 1B*). This finding suggests that the specific dynamics of enhancer activation cannot be readily distinguished by the presence or absence of these TF-binding motifs alone, and suggests that the sequence features that determine whether an enhancer is constitutively active or signal-responsive are more complex and remain to be identified (*Bevington et al., 2016*; *Comoglio et al., 2019*).

Since these extended AP-1 and TEAD motifs were defined by their enrichment at enhancers in the C57BL/6 J genome (in comparison to GC-matched control regions), we next sought to determine the impact of SNPs within these motifs on AP-1 and TEAD binding at active enhancers using allele-specific TF-binding data (*Figure 3A–B*). We performed a series of validations to ensure that distinct methods (ChIP-seq and CUT&RUN) were providing similar quantitative information on TF-binding levels (*Figure 3—figure supplement 2A-G*). We reasoned that an increased frequency of SNPs in allele-specific enhancers would occur only at nucleotides required for sequence-dependent binding of these TFs and not at neighboring regions flanking these nucleotides (*Maurano et al., 2015*). For active enhancer loci with allele-specific Fos binding, an enrichment of SNPs is observed at the core AP-1 motif (*Figure 3C*; n=263 allele pairs). Within the core motif, the lowest enrichment of SNPs was observed at the central nucleotide, consistent with in vitro experiments suggesting that this nucleotide does not strongly influence AP-1-binding affinity (*Risse et al., 1989*). Recent in vitro studies of AP-1 binding affinity to AP-1/TRE motifs suggests that the three nucleotides flanking the core AP-1 motif (TGASTCA) can strongly modulate AP-1 TF binding by altering the shape of the AP-1/TRE motif (*Leonard and Kerppola, 1998*; *Rohs et al., 2010*; *Yella et al., 2018*). Given these data, we assessed whether these flanking sequences play a role in determining AP-1 binding site selection in chromatin. Our data in *Figure 3C* suggests SNPs at the 5' and 3' flanking nucleotides of the AP-1/TRE motif (VTGACTCAB) can affect AP-1 binding at active Fos-bound enhancers. More broadly, there is an enrichment of SNPs in the three nucleotides flanking each side of the core AP-1 binding site when considering all allele-specific Fos-bound sites from our data (NNVTGACTCABNN; 9.9% and 5.6% at allele-specific and shared Fos peaks, respectively; *Figure 3—figure supplement 1C*). These results provide further evidence that sequences immediately flanking core AP-1 motifs should be considered in future assessments of AP-1 binding motif preferences.

Next, we found that an additional nucleotide beyond the core TEAD-binding site (GGAAT) was restricted to G/T in the KMAC output (GGAATK), and SNPs were enriched at all positions within this motif at allele-specific Tead1-bound enhancers (*Figure 3D*). For allele-specific CTCF sites with >2-fold difference in signal, we found a~14 bp window of enriched SNPs (i.e. broader than typical DNA-binding TFs, like AP-1 or TEAD) that disrupt CTCF binding (*Figure 3E*), closely mirroring the ~15–20 bp sequence that CTCF is predicted to bind in vivo (*Kim et al., 2007*). These data indicate that AP-1 and TEAD motifs are the most enriched TF motifs within active enhancers in fibroblasts and functionally validate the importance of motif-flanking nucleotides for TF binding in the native chromatin context.

We next sought to use our allele-specific TF-binding data to perform a targeted analysis of how an allele-specific loss in AP-1 or TEAD occupancy impacts enhancer chromatin state. We separately

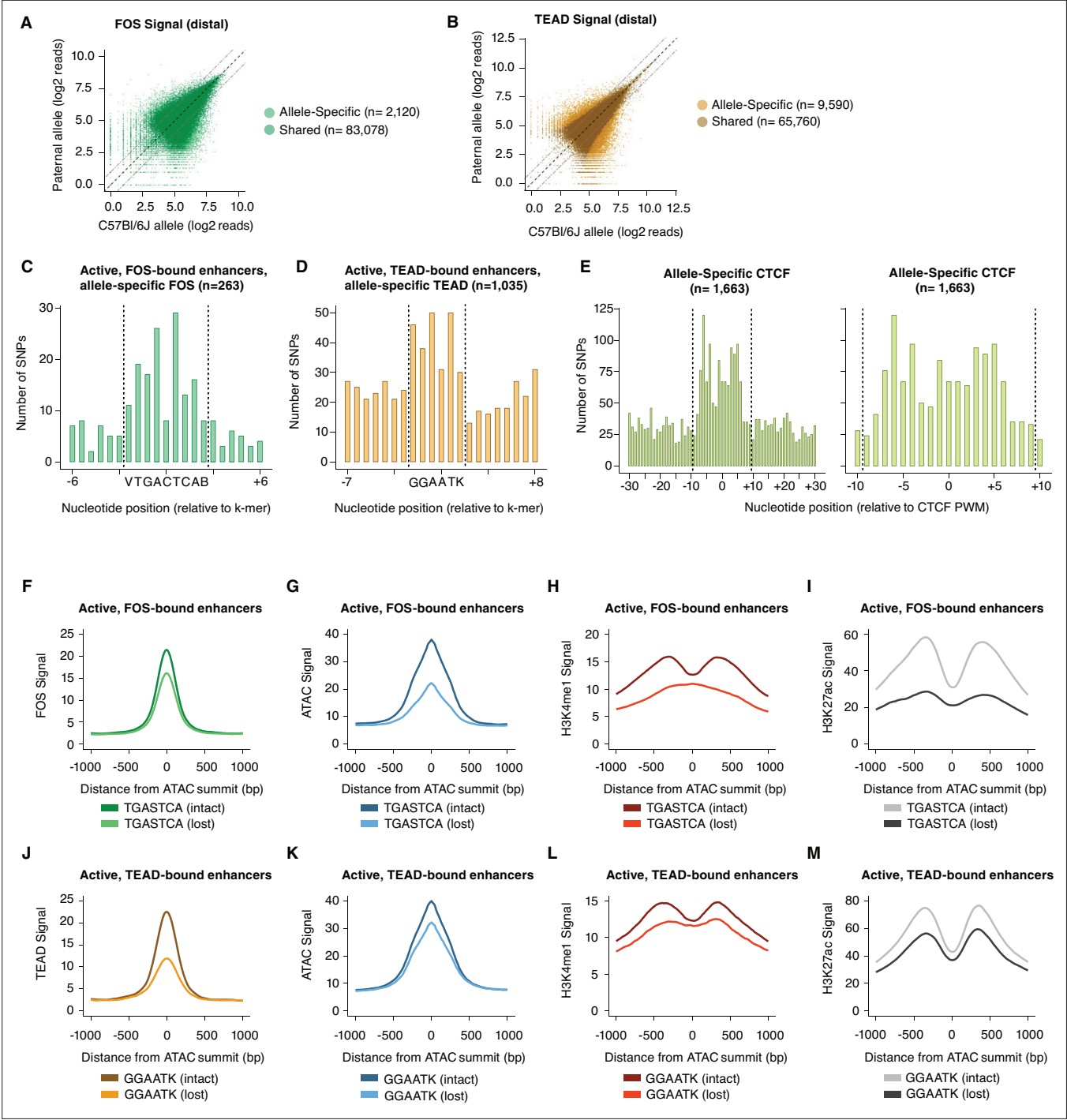

**Figure 3.** Sequence motifs and changes in chromatin state at allele-specific TF-bound sites. (**A–B**) Scatterplots of maternal (C57BL/6 J) and paternal allele-specific signal for AP-1 and TEAD binding (n=85,198 distal Fos, and 75,350 distal Tead1 allele pairs, respectively). Points indicated in light and dark colors represent peaks with and without a significant skew in signal between alleles, respectively (FDR <0.1 with DESeq2). (**C–D**) Distribution of SNPs centered on respective k-mers (denoted by dashed lines) at allele-specific, active, and gene-distal Fos and Tead1 peaks with >2-fold difference in binding signal across alleles (n=263 and n=1035 allele pairs, respectively). (**E**) Distribution of SNPs centered on CTCF PWM (JASPAR matrix ID MA0139.1; denoted by dashed lines) at allele-specific, gene-distal CTCF peaks with >2-fold difference in binding signal across alleles (n=1,663 allele pairs). (**F–I**) Aggregate plot of allele-specific Fos, ATAC-seq, H3K4me1, and H3K27ac reads centered on ATAC-seq summits at active Fos peaks. These sites have been selected because they contain a single SNP/indel-containing AP-1 site and no shared AP-1 sites within 75 bp of the ATAC-seq summit. Signal is compared between alleles with intact versus mutated core AP-1 motifs (TGASTCA; n=1307 allele pairs). (**J–M**) Aggregate plot of allele-specific Tead1, ATAC-seq, H3K4me1, and H3K27ac reads centered on ATAC-seq summits at active Tead1 peaks. These sites have been selected because they

_Figure 3 continued on next page_

*Figure 3 continued*

contain a single SNP/indel-containing TEAD site and no shared TEAD sites within 75 bp of the ATAC-seq summit. Signal is compared between alleles with intact versus mutated core TEAD motifs (GGAATK; n=1132 allele pairs).

The online version of this article includes the following figure supplement(s) for figure 3:

**Figure supplement 1.** Enriched DNA-binding motifs at active and constitutive enhancers.

**Figure supplement 2.** Validation of Fos and H3K27ac CUT&RUN experiments.

identified a set of active enhancers that have a single instance of their core motifs (TGACTCA or GGAAT) and have a SNP/indel that alters this binding motif into a sequence not predicted to bind AP-1 or TEAD based on in vitro binding experiments, respectively. For both classes of TFs, motif-disrupting SNPs are correlated with a marked loss of binding of their cognate TFs, as expected, but this incomplete loss (in aggregate) also suggests that SNPs within core motifs alone are not completely predictive of changes in AP-1 or TEAD binding (*Figure 3F and J*). Loss of AP-1 binding is associated with a substantial decrease in chromatin accessibility, H3K4me1/2, and H3K27ac levels on the allele with the mutated AP-1 site (*Figure 3G–I*), consistent with our previous observations from a smaller set of enhancers (*Vierbuchen et al., 2017*). This finding suggests that at enhancers that contain a single consensus AP-1 site and are bound by Fos/Jun, a variant that changes a nucleotide in the core AP-1 motif is likely to result in a complete loss of enhancer function, consistent with data from a smaller set of plasmid-based reporters that suggest AP-1 motifs are required for transcriptional activation (*Malik et al., 2014*; *Liu et al., 2016*). Similar analysis of Tead1-bound enhancers revealed a more modest decrease in ATAC-seq, H3K4me1/2, and H3K27ac signal associated with the allele lacking a predicted TEAD motif (*Figure 3K–M*), suggesting that loss in TEAD occupancy has less severe consequences on enhancer function than loss of AP-1 binding. Nevertheless, these data suggest that both AP-1 and TEAD motifs play a central role in enhancer function in fibroblasts.

## AP-1 TFs facilitate binding of TEAD TFs to enhancers

Although SNPs that disrupt core TF-binding motifs (AP-1, TEAD, and ETS) are enriched at enhancers with allele-specific TF binding, our data also indicate that SNPs in these motifs are not sufficient to explain all instances of functional variation between enhancer alleles. For example, among all enhancers in the top decile of allele-specific H3K27ac signal, only 13.3% had a SNP/indel overlapping a core AP-1, TEAD, and/or ETS motif in their central region. In contrast, we found that 21.5% of allele-specific CTCF binding sites (with >2-fold difference in CTCF signal) that contain a CTCF motif (identified using position weight matrix match) had at least one SNP/indel overlapping the CTCF binding site. These data favor a model in which other types of SNPs outside core TF-binding motifs can collectively modulate enhancer activity.

In previous work, we found that strain-specific instances of AP-1 TF binding in MEFs (in a comparison of two inbred mouse strains) that lack a mutation in a core AP-1 site were enriched for SNPs in TEAD motifs, suggesting a model in which AP-1 binding was dependent, at least in part, on the presence of TEAD binding sites (*Vierbuchen et al., 2017*). However, we lacked TEAD binding data, which prevented us from examining in depth the sequence determinants and functional relationship of AP-1 and TEAD binding at enhancers across the genome. Other data have suggested that AP-1 and TEAD TFs coordinately regulate transcriptional programs critical for cell growth and proliferation during normal development and in the context of cancer (*Liu et al., 2016*; *Zanconato et al., 2015*; *Park et al., 2020*; *He et al., 2021*). Since multiple AP-1 and TEAD TFs are also often co-expressed in the same cell types and can play functionally redundant roles with one another (*Seo et al., 2021*), it has been difficult to examine how these two TFs that exhibit extensive co-occupancy work together at enhancers to regulate gene transcription. With our newly generated AP-1 and TEAD binding data across four wild-derived inbred $F_1$-hybrid lines, we could more systematically examine a larger number of loci to define the functional relationship between AP-1 and TEAD.

We first quantified how often consensus TF motifs are mutated at allele-specific versus shared AP-1 and TEAD peaks. If the binding of a given TF was entirely dependent on the occupancy of another TF, we would expect to observe a similar loss in binding of the dependent TF, regardless of which TF motif was mutated. For these analyses, we included all distal Fos and Tead1 peaks in our dataset, including those that do not co-occur with H3K27ac. We observed that AP-1 motif mutations

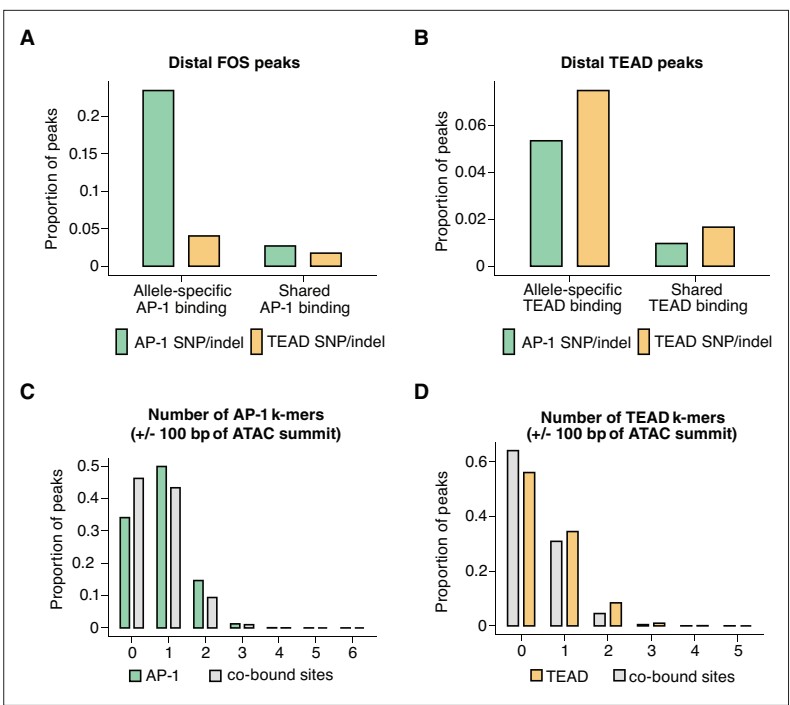

**Figure 4.** Mechanisms of hierarchical binding for AP-1 and TEAD TFs. (**A**) Percentage of allele-specific (n=1635 allele pairs) versus shared (n=142,778 allele pairs) gene-distal Fos peaks that contain strain-specific core AP-1 (TGASTCA; Fisher's exact test, p<2.2 x 10⁻¹⁶) or extended TEAD (GGAATK; Fisher's exact test, p=2.123 x 10⁻⁹) k-mer(s) within 75 bp of their respective ATAC-seq summits. (**B**) Percentage of allele-specific (n=9416 allele pairs) versus shared (n=65,934 allele pairs) gene-distal Tead1 peaks that contain strain-specific core AP-1 (TGASTCA; Fisher's exact test, p<2.2 x 10⁻¹⁶) or extended TEAD (GGAATK; Fisher's exact test, p<2.2 x 10⁻¹⁶) k-mer(s) within 75 bp of their respective ATAC-seq summits. (**C**) Percentage of AP-1-only (n=15,709 loci) peaks versus AP-1/TEAD co-bound peaks (n=2797 loci) in the C57BL/6 J genome with at least one bindable AP-1 k-mer (VTGACTCB, VTGAATCAB, or VTTAGTCAY) present within 50 bp of their respective ATAC-seq summits (Fisher's exact test, p<2.2 x 10⁻¹⁶). (**D**) Percentage of TEAD-only (n=2541 loci) peaks versus AP-1/TEAD co-bound peaks in the C57BL/6 J genome with at least one extended TEAD k-mer (GGAATK) present within 50 bp of their respective ATAC-seq summits (Fisher's exact test, p=2.406 x 10⁻⁹).

are frequently associated with a loss of TEAD binding, whereas AP-1 binding is more weakly affected by TEAD motif mutations (*Figure 4A–B*). Strikingly, AP-1 motif mutations were as enriched at allele-specific TEAD peaks as TEAD mutations were (compared to sites with shared TEAD binding). Analysis of AP-1 and TEAD co-bound sites (independent of whether they contained consensus AP-1 or TEAD motifs) further supported a hierarchical binding relationship between these TFs. For example, 50.2% (n=821/1635 allele pairs) of allele-specific Fos-bound sites also exhibit an allele-specific loss of Tead1 binding, whereas only 8.7% (n=821/9416 allele pairs) of allele-specific Tead1 peaks showed significant allele-specific Fos signal. In summary, these data are consistent with previous studies that suggest that AP-1 can serve as a pioneer TF to facilitate the binding of other TFs, such as the glucocorticoid receptor, PU.1, and C/EBP (*Biddie et al., 2011*; *Heinz et al., 2013*), and that AP-1 binding is required for inducible chromatin remodeling and nucleosome displacement at late-response gene enhancers in fibroblasts (*Vierbuchen et al., 2017*).

Data from in vitro studies examining TF-binding specificity have shown that TFs that bind to composite motifs often prefer sequences that are distinct from their consensus individual motifs (*Jolma et al., 2015*). This led us to consider the possibility that sites at which AP-1 and TEAD bind together might exhibit differential motif requirements from sites where only one of these two TFs bind. We observed that AP-1-only peaks contain at least one AP-1 k-mer found using KMAC (65.9%; VTGACTCAB, VTGAATCAB, or VTTAGTCAY), whereas AP-1/TEAD co-bound peaks were less likely to contain a consensus motif (53.8%; *Figure 4C*). Similarly, TEAD-only peaks (44.0%) had a higher frequency of TEAD k-mers identified with KMAC (GGAATK) than AP-1/TEAD peaks (36.0%;

*Figure 4D*). These data suggest that the motif requirements for AP-1/TEAD co-bound regions are slightly more flexible than sites at which only one of the TFs bind.

## Identification of sequence features that determine AP-1/TEAD co-binding at enhancers

Thus far, our data suggest that many instances of AP-1 and TEAD binding cannot be explained solely by mutations in consensus, core motifs for these TFs. This lack of enriched TF motif mutations has been observed for other classes of TFs and in a variety of model systems, suggesting that this is a general, unresolved problem in genetic studies of TF binding (*Deplancke et al., 2016*). Our dataset allowed us to systematically look for recurrent features of SNPs/indels associated with allele-specific AP-1 or TEAD binding outside of core motifs for these TFs. These analyses can help reveal additional sequence motifs that influence AP-1 and TEAD binding, such as binding sites for other TFs that bind together with AP-1 or TEAD to establish chromatin accessibility. In particular, SNPs outside known TF-binding sites allow us to dissect the role of motif spacing on the ability of TFs to cooperate with one another to bind enhancers. Subtle changes in motif syntax have been shown to alter enhancer function (*Erceg et al., 2014*; *Farley et al., 2016*; *Shen et al., 2021*), and conversely, the arrangement of TF-binding motifs can also be highly flexible in other contexts (*Arnosti and Kulkarni, 2005*; *Junion et al., 2012*; *King et al., 2020*; *Jindal and Farley, 2021*).

First, we identified allele-specific gene-distal binding sites for Fos, Tead1, and CTCF, and then examined the frequency of SNPs/indels (relative to the ATAC-seq peak center) at these sites compared to sites with shared binding on both alleles. In *Figure 2I*, we plotted SNP/indel distributions at enhancers with allele-specific histone acetylation, whereas these analyses focus instead on TF-binding

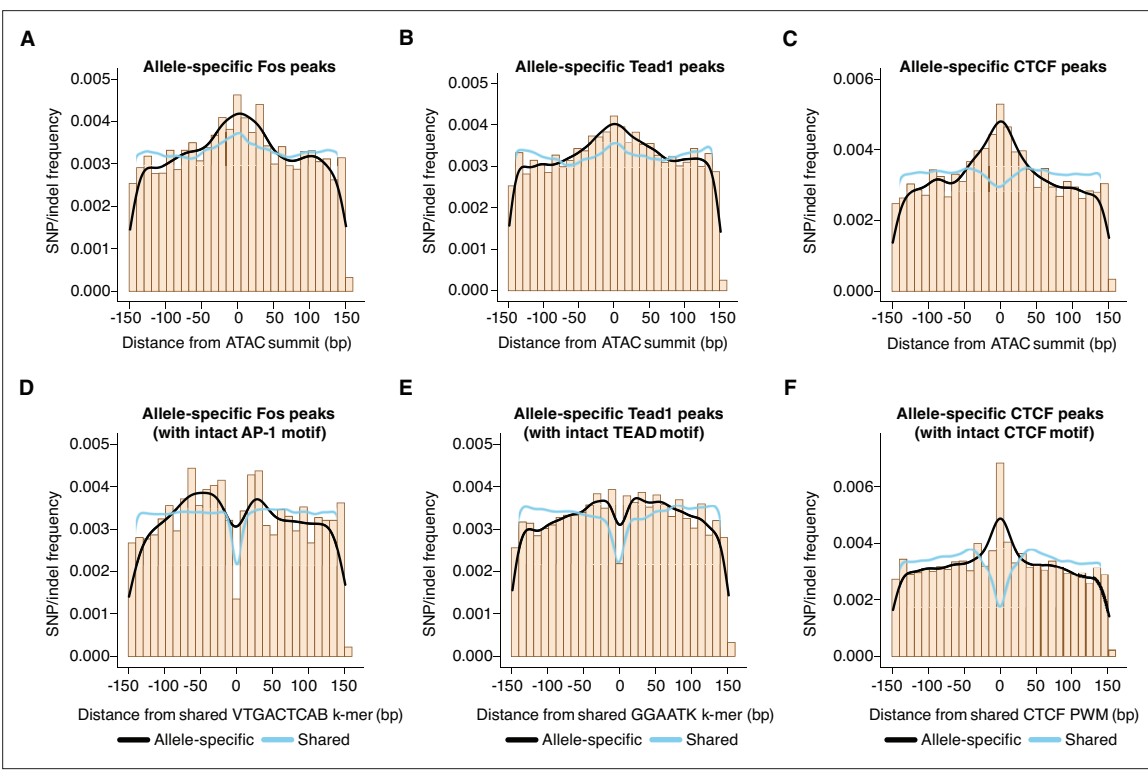

**Figure 5.** Distribution of genetic variants that influence AP-1, TEAD, and CTCF binding. (**A–C**) Frequency of SNPs/indels at positions relative to ATAC-seq summits at allele-specific (with >2-fold difference in signal between alleles) versus shared gene-distal Fos, Tead1, and CTCF peaks (Pearson's Chi-squared test, p=9.7 x 10⁻⁸ for AP-1, p<2.2 x 10⁻¹⁶ for TEAD, p<2.2 x 10⁻¹⁶ for CTCF, 100 bp window centered on ATAC-seq summit in all cases). (**D–E**) Frequency of SNPs/indels at positions relative to shared VTGACTCAB and GGAATK k-mers within 75 bp of the ATAC-seq summit at allele-specific (with >2-fold difference in signal between alleles) versus shared gene-distal Fos and Tead1 peaks, respectively. Sites have been filtered to exclude any peaks that include SNPs/indels that overlap their cognate k-mers. (**F**) Frequency of SNPs/indels at positions relative to shared CTCF PWM (MA0139.1) within 75 bp of the ATAC-seq summit at allele-specific (with >2-fold difference in signal between alleles) versus shared gene-distal CTCF peaks. Sites have been filtered to exclude any peaks that include SNPs/indels at disrupt the CTCF PWM in a strain-specific manner.

sites independent of H3K27ac levels. When comparing allele-specific and shared TF peaks, we found an increased frequency of SNPs/indels within an ~100 bp window centered on the ATAC-seq peak summit, which is similar to the pattern observed at enhancer loci with allele-specific H3K27ac levels (*Figure 5A–C*).

Next, given that AP-1 motif SNPs likely contribute to the distribution observed in *Figure 5A*, we repeated this analysis, but excluded allele-specific AP-1 peaks that have an SNP/indel in their extended AP-1 motif (VTGACTCAB) and plotted the SNP/indel frequency relative to this motif instead of the ATAC-seq peak summit. This revealed an enrichment of SNPs/indels within +/-~50 bp of the AP-1 motif (*Figure 5D*). SNPs/indels were similarly distributed relative to TEAD motifs at TEAD peaks (*Figure 5E*). These observed patterns of SNPs/indels are consistent with a collaborative competition model for AP-1 and TEAD binding. The collaborative competition model provides an explanation for how TFs gain access to enhancer sequences that form nucleosomes. In this model, simultaneous binding of multiple TFs is thought to be essential to outcompete high-affinity interactions of histone octamers with these enhancer DNA sequences. Biophysical experiments suggest that these collaborating TFs must bind to the same half of the nucleosome to compete against the histone octamer for binding (i.e. the same side relative to the nucleosome dyad, <75 bp from one another; *Miller and Widom, 2003*; *Moyle-Heyrman et al., 2011*).

In contrast with the pattern of SNPs observed at allele-specific AP-1 and TEAD peaks, when we plotted the distribution of SNPs/indels at allele-specific and shared CTCF peaks, we observed a narrow enrichment of SNPs (within +/- 10 bp) relative to the CTCF motif at allele-specific compared to shared peaks (*Figure 5F*). This result indicate a more restricted length scale at which genetic variants can disrupt CTCF binding than those that we observed for Fos and Tead1, and suggests that CTCF binding is less dependent on binding of additional, collaborating TFs.

## Contribution of partial or degenerate AP-1 motifs to AP-1 binding affinity

Binding of TFs to their cognate motifs on nucleosomes is often restricted by steric hindrance between TFs and histone octamers. In particular, some pioneer TFs are thought to preferentially bind partial motifs over full motifs on nucleosomes (*Soufi et al., 2015*; *Roberts et al., 2021*). We considered the possibility that some instances of allele-specific AP-1 binding where there is an absence of a core AP-1 motif mutation could be explained by SNPs in nearby partial or degenerate TF-binding motifs not readily detected by traditional searches.

To examine whether binding to AP-1 half sites contributes to AP-1 recognition at enhancers, we chose to examine the frequency of TGASVDB k-mers at AP-1 bound sites. It should be noted that this motif is able to identify AP-1 half sites, and at the same time, capture degenerate or low-affinity AP-1 motifs that are difficult to detect from traditional motif searches because they bear little resemblance to predicted core motifs (*Kribelbauer et al., 2019*). Allele-specific and shared AP-1 peaks contained, on average, a similar number of AP-1 half sites (in the context of TGASVDB motifs; mean = 1.17 and 1.20 occurrences per peak in the central 150 bp, respectively). However, we observed a ~2-fold greater frequency of AP-1 half sites containing SNPs (TGASVDB) in allele-specific versus shared Fos peaks (mean = 0.25 and 0.13 occurrences per peak, respectively), suggesting that AP-1 half sites contribute to AP-1 TF binding in chromatin. However, based on our prior analysis of mutations in full AP-1 sites, it is clear that disruption of one of two half sites within an AP-1 consensus motif has a strong effect on AP-1 binding in most cases, which suggests that AP-1 half sites alone might not be sufficient for binding without another intact AP-1 motif at the same enhancer (*Figure 3C and F*). Thus, we favor a model in which AP-1 half sites play an accessory role in modulating levels of AP-1 occupancy, and are unlikely to be sufficient for AP-1 binding by themselves in the absence of a full AP-1 motif.

## Identification of k-mers predictive of AP-1 binding and/or activity using machine learning

Since core TF-binding motifs alone cannot fully distinguish TF-bound alleles from those in which TFs are bound, we reasoned that additional k-mers could contribute to our ability to distinguish TF-bound sites versus non-bound sites in the C57BL/6 J genome, and we considered the possibility that identification of these k-mers might help identify other motifs that are recurrently mutated in our

allele-specific TF-binding data. Therefore, we applied a gapped k-mer SVM approach (gkm-SVM) to our datasets that has been optimized to detect k-mers of similar length to typical TF-binding motifs (*Ghandi et al., 2016*). Support vector machine (SVM) algorithms have been utilized in a variety of contexts to perform classification of DNA sequences in a supervised manner (*Barozzi et al., 2014*; *Ghandi et al., 2016*; *Van den Bosch et al., 2022*). It should be noted that k-mers identified by gkm-SVM are simply DNA sequences of k length that can discriminate two sets of input sequences, and do not necessarily correspond to TF-binding sites per se.

We first compared 60 bp of DNA sequences from AP-1 peaks in C57BL/6 J MEFs (positive set) to GC- and length-matched, randomly sampled background DNA sequences from the C57BL/6 J genome (negative set). The area under the receiver operating characteristic curve (AUROC = 0.872) from this gkm-SVM analysis is highly similar to the corresponding value obtained from a control analysis of CTCF peaks from human cells, suggesting that the gkm-SVM is able to classify Fos-bound and unbound regions with a low rate of detecting false positives while correctly assigning true negatives. Similarly, a relatively high value for the area under the precision-recall curve (AUPRC = 0.881) indicates that the gkm-SVM is able to reliably distinguish true and false positives. Together, these results suggest that the information within the central 60 bp sequences (+/-30 bp relative to the ATAC-seq summit) at Fos-bound peaks is sufficient to train a model to reliably distinguish Fos-bound sites from control non-coding regions of the same genome (*Figure 6—figure supplement 1A*). Inclusion of additional sequence beyond this central 60 bp (up to a total of 300 bp) had only a slight positive effect on the performance of the model (*Figure 6A–B*). Conversely, shortening of DNA sequence below 60 bp resulted in a drop-off in performance of the model. k-mers containing AP-1 sites were the largest contributors to the performance of the model, as expected (*Figure 6—figure supplement 1D*). Next, to determine whether sequences outside of the AP-1 motif contribute to the performance of the model, we repeated this same analysis, but we computationally masked all occurrences of core AP-1 sites. This revealed a slight drop in AUROC (unmasked = 0.874, masked = 0.804) and AUPRC (unmasked = 0.884, masked = 0.794) values, suggesting that the model retains some predictive capacity when core AP-1 motif sequences are excluded (*Figure 6A–B*, *Figure 6—figure supplement 1E*).

Interestingly, k-mers containing AP-1 sites also contributed the most to model performance when gkm-SVM was applied to Tead1-bound sites (*Figure 6—figure supplement 1B and F*), consistent with our observations that AP-1 binding is required for TEAD binding at many enhancers (*Figure 5*). When we ran the gkm-SVM on CTCF peaks, we observed a highly distinct set of enriched k-mers from those found at AP-1 peaks. Many of these identified k-mers matched the well-documented CTCF binding site, as expected (*Figure 6—figure supplement 1C and G*).

We next sought to apply this gkm-SVM approach to attempt to identify k-mers that distinguish between AP-1 binding sites with and without H3K27ac. Our data indicate that the AP-1 binding is required for the function of many of the active enhancers at which they bind in MEFs. However, AP-1 binding alone is clearly not sufficient for enhancer activity. For example, 34.9% of gene-distal Fos peaks do not overlap H3K27ac peaks. This suggests that the sequence features that are permissive for AP-1 binding in MEFs might be separable from those that confer activity. For this gkm-SVM analysis, we selected a curated set of Fos-bound allele pairs (n=2,697) that (1) have equivalent levels of Fos binding, (2) contain a consensus AP-1 site on both alleles, but (3) exhibit allele-specific H3K27ac levels. We input 60 bp DNA sequences (centered on the shared AP-1 consensus motif) from the active (positive set) and inactive (negative set) alleles at Fos-bound enhancers (*Figure 6—figure supplement 1H*). In contrast to the results above, the gkm-SVM failed to discriminate between these two sets of sites (AUROC = 0.086 and AUPRC = 0.318), suggesting that sequence features predictive of H3K27ac are more complex and cannot be readily captured by this k-mer based SVM approach.

## Generalizability of sequence determinants of AP-1 binding across cell types and species

Having defined some features of sequences that determine AP-1 binding to CREs in MEFs, we next extend our analyses to data derived from a larger number of other cell types. To do this, we used DNase-seq footprinting data generated from a large panel of human tissues and cell types (*Vierstra et al., 2020*). These data provide an unbiased view of individual TF-DNA binding interactions within CREs.

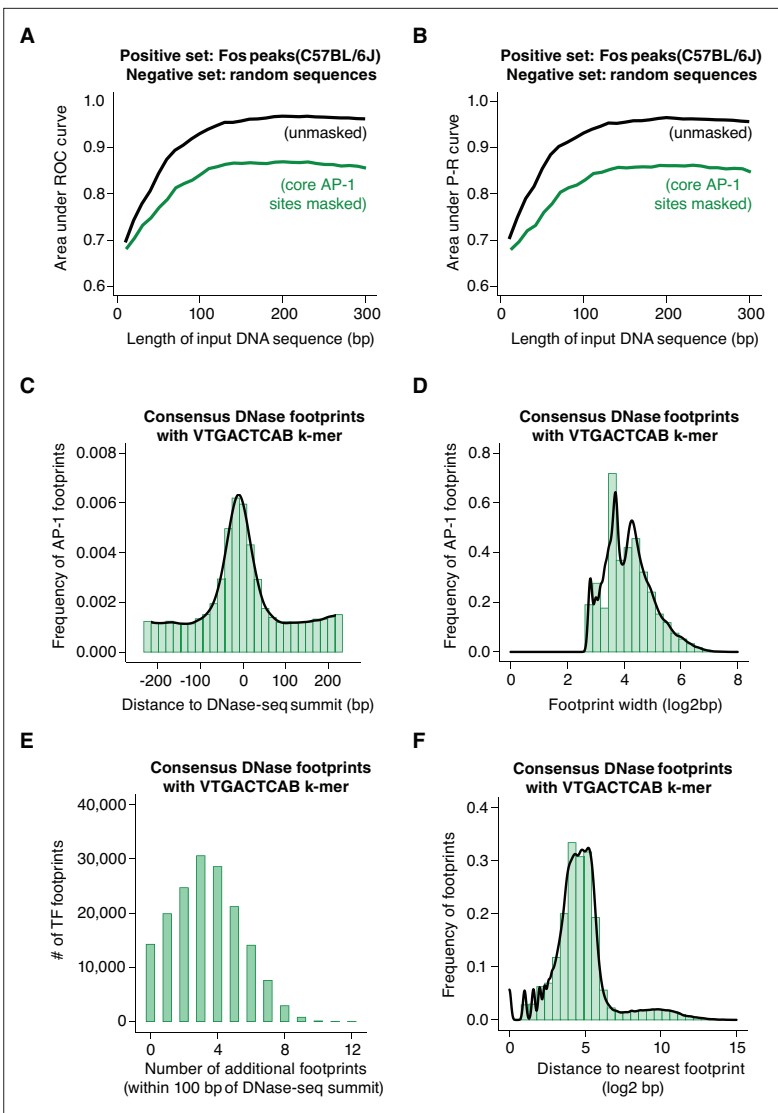

**Figure 6.** Machine learning prediction of AP-1 binding sites genome-wide. (**A–B**) Area under ROC and P-R curves for gkm-SVM comparison of Fos peaks in the C57BL/6 J genome (positive set) and length-matched, random genomic regions (negative set). Shown are AUC values for different lengths of input DNA sequence, ranging from 10 to 300 bp, indicated in black. The same analysis was repeated after masking all instances of core AP-1 motifs (TGASTCA; n=4000 randomly selected loci), indicated in green. (**C**) Frequency of consensus, human DNase footprints (from *Vierstra et al., 2020*) containing an extended AP-1 k-mer (VTGACTCAB) at positions relative to DNase-seq summits (n=164,705 footprints). (**D**) Width of DNase footprints that contain an extended AP-1 k-mer (VTGACTCAB). (**E**) Number of additional DNase footprints within 100 bp of DNase-seq summits at DNase peaks with a VTGACTCAB-containing footprint. (**F**) Distance between VTGACTCAB-containing footprints and nearest neighboring DNase footprint.

The online version of this article includes the following figure supplement(s) for figure 6:

**Figure supplement 1.** gkm-SVM analysis of Fos, Tead1, and CTCF peaks.

First, we identified TF footprints that overlap an extended AP-1 motif (VTGACTCAB) within CREs, which we interpret as individual instances of AP-1 binding. We found a total of 164,705 TF footprints (from among >4 × 10⁶ total footprinted regions) that contain VTGACTCAB motifs. These AP-1 footprints were centrally enriched within CREs (*Figure 6C,*) consistent with the distribution of AP-1 k-mers observed at AP-1 bound peaks in MEFs, as well as previous data examining AP-1 motif frequency within human DNase-seq peaks (*Grossman et al., 2017*). The majority of AP-1 footprints were <30 bp in width (83.3%; median = 17 bp), which suggests that they represent the footprint caused by binding

of a single AP-1 homo/heterodimer (*Figure 6D*). CREs with AP-1 footprints typically have a total of ~3–4 additional TF footprints (*Figure 6E*), and the median distance between AP-1 footprints and the nearest other TF footprint is ~24 bp (*Figure 6F*). gkm-SVM analysis of sequences flanking AP-1 footprints (60 bp windows) revealed an enrichment of TEAD and ETS k-mers, consistent with our observations at AP-1 bound sites in MEFs (*Figure 6—figure supplement 1I*). Together, these data suggest a model in which AP-1 is critical for CRE function across many cell types and provide further insight into the nature of TF-binding events that occur with AP-1 binding at CREs active across cellular contexts, such as TEAD and ETS. These data will be valuable for disentangling the complex sequence features that control AP-1 binding and enhancer function across diverse cell types and tissues.

## Discussion

In this study, we leverage natural genetic variation across inbred mouse strains to identify sequence variants associated with differential TF binding and/or enhancer activity in their endogenous genomic context. To systematically assess the effect of many genetic variants on CRE function, we mapped TF binding (AP-1, TEAD, CTCF) and multiple chromatin features (ATAC-seq, H3K27ac, H3K4me1/2, H3K4me3) in up to ten distinct alleles for each CRE locus. By assessing the frequency and distribution of genetic variants at large numbers of CREs with shared or allele-specific TF binding and/or *cis*-regulatory activity, we define features of *cis*-acting genetic variants that are most predictive of differences in chromatin state and/or TF binding.

We find from our analysis of enhancer alleles with different H3K27ac levels that loss of the active enhancer histone modification H3K27ac is generally not genetically separable from loss of H3K4me1/2. This is interesting to consider given a number of previous observations about the relative contribution of these histone modifications to enhancer function: (1) we previously found that enhancers that regulate late-response genes exhibit H3K4me1 enrichment in serum-starved MEFs, but have low chromatin accessibility and lack H3K27ac, and upon serum-stimulation, gain H3K27 acetylation and inducibly bind AP-1 TFs (*Vierbuchen et al., 2017*), (2) recent studies suggest that enhancers with H3K4me1 enrichment that lack H3K27ac are not, in fact, poised for future activity, but instead that this chromatin state is a remnant of activity in a recent, prior developmental stage (*Kim and Shiekhattar, 2015*; *Jadhav et al., 2019*), and (3) catalytic mutants of Mll3/4, enzymes responsible for H3K4me1/2 deposition, do not appear to affect recruitment of RNA polymerase II to enhancers suggesting that the H3K4me1/2 modification is not required for enhancer function (*Dorighi et al., 2017*; *Jang et al., 2017*; *Rickels et al., 2017*). Taken together, these observations suggest that in a given cell type or context, H3K4me1/2-only enhancers might exhibit different *cis*-regulatory features compared to enhancers that have H3K27ac enrichment because they represent enhancers that were active in a previous developmental stage characterized by a distinct complement of TFs expressed.

The functional sequence variants between inbred mouse strains that we identified provide insight into enhancer turnover that occurs across evolution (*Villar et al., 2015*). We find that greater numbers of SNPs/indels at enhancers are correlated with higher probabilities of allele-specific enhancer activity, and that allele-specific enhancers in fibroblasts are also less conserved across species than shared enhancers. Enhancers are thought to turn over rapidly because many loci contain multiple enhancers with overlapping *cis*-regulatory activity, such that loss of any individual single enhancer is often insufficient to cause a large change in gene expression and/or result in an organismal phenotype (*Osterwalder et al., 2018*). It would also be interesting to examine how the impact of SNPs on enhancer activity correlates with their frequency in natural populations or whether the SNP represents a derived or ancestral state across the broader rodent lineage, as SNPs with larger impacts on enhancer function would be expected to be found at lower frequencies if they are potentially deleterious to overall fitness.

We also found that SNPs/indels that are associated with allele-specific H3K27ac levels tend to occur within ~50 bp of the center of the accessible chromatin region used to define enhancer sequences. This is further supported by enhancer loci in which alleles differ by a single SNP/indel. At such enhancer alleles, these single sequence variants are almost certainly causal and thus likely to explain observed differences in enhancer activity. Together, these data suggest that SNPs within the central region of enhancers should be prioritized in genome-wide association studies for human traits and/or disease risk.

The distribution of SNPs that impact AP-1 binding to enhancers is interesting to consider in the context of a recent paper that looked at sequence features that distinguish AP-1 bound enhancers with high versus low activity in reporter assays (*Chaudhari and Cohen, 2018*). Their analysis using a supervised machine learning approach (gkm-SVM) suggests that most of the variation in AP-1 bound enhancer activity can be predicted using input sequences that consist of the AP-1 core motif and an additional +/-10 bp on each side. The close proximity of these sequences to AP-1 sites contrasts with the broader window (+/- 50 bp relative to AP-1 motifs) that we observe to be important for determining AP-1 binding to chromatin in fibroblasts. More generally, these observations suggest that the sequence determinants of enhancer activity might be more complex within the genomic context than what has been observed in reporter assays. In addition, we found that the gkm-SVM has limited ability to predict activity levels of AP-1 bound sites, suggesting that deep learning approaches might be required to delineate higher order sequence features associated with active enhancers in a given cellular context (*Avsec et al., 2021*; *de Almeida et al., 2022*).

The observed requirement for AP-1 for TEAD binding to enhancers is interesting to consider given that these two distinct TFs have been previously shown to co-regulate gene expression programs associated with cell proliferation and tissue growth. Both AP-1 and TEAD are transcriptional effectors of intercellular signaling pathways. AP-1 TFs are activated by Ras/MAPK signaling and TEAD TFs are required for the binding of the transcriptional co-activators YAP and TAZ, whose nuclear localization is directly regulated by Hippo signaling. Our data suggests a mechanism for crosstalk between these two signaling pathways in which the transcriptional output of the Hippo pathway can be modulated depending on whether Ras/MAPK is active or not. This instructive function of AP-1 in selecting the enhancers at which TEAD TFs can bind is similar to the role for AP-1 in facilitating the binding of transcriptional effectors from several other signaling pathways, including the glucocorticoid receptor, NF-κB, and SMAD (*Biddie et al., 2011*; *Heinz et al., 2013*; *Li et al., 2019*).

Enhancer sequences tend to be occluded by histone octamers prior to TF binding, suggesting that during the process of enhancer selection, some TFs must be able to bind to their cognate motifs in a nucleosomal context (*Tillo et al., 2010*; *Barozzi et al., 2014*). This complicates efforts to determine sequence features required for TF binding because the affinity of a TF for nucleosomal and naked DNA often differ significantly. Furthermore, different classes of TFs utilize distinct mechanisms to engage with their cognate motifs on nucleosomes (*Michael and Thomä, 2021*). Based on in vitro nucleosome-binding studies, bZIP TFs, which include AP-1 TFs, can only bind to their cognate motifs on nucleosomes when their motifs are present on outer regions of nucleosomes (i.e. furthest from predicted dyad locations; *He et al., 2013*). These outer regions are thought to be more accessible for TF binding because they are intermittently unwound from the histone octamer (known as nucleosome breathing; *Zhu et al., 2018*; *Zhou et al., 2019*). This is consistent with structural data suggesting that AP-1 TFs cannot bind their full cognate motif (TGASTCA) on nucleosomes due to steric constraints (*Michael and Thomä, 2021*). We observe an enrichment of SNPs in AP-1 half sites (TGASVDB) at allele-specific AP-1 peaks, raising the possibility that partial AP-1 motifs contribute to AP-1 binding to nucleosomal enhancers. The binding to partial motifs has been observed for other nucleosome-binding TFs, such as OCT4 (*Soufi et al., 2015*). Thus, we favor a model in which (1) AP-1 recognizes full motifs towards the edges of nucleosomes, (2) AP-1 dimers can then bind both halves of the core AP-1 site when it is accessible upon nucleosome breathing, (3) the initial binding of AP-1 facilitates the binding of other dependent TFs (e.g. TEAD) via collaborative competition to evict the histone octamer and/or recruit co-regulatory proteins and chromatin remodelers to enhancers, and (4) AP-1 might be able to bind half motifs absent nucleosome breathing at any position on the nucleosome. In future, fully delineating sequence requirements for AP-1 binding will require detailed in vitro and structural experiments using naturally occurring enhancer sequences, as well as deep learning approaches applied to genomic data of AP-1 binding from multiple cell types and genotypes (*Avsec et al., 2021*).

Our $F_1$-hybrid dataset has provided new insights into how DNA sequences within CREs contribute to TF binding and enhancer function. We believe that this $F_1$-hybrid approach for examining TF function is a powerful tool to uncover sequence determinants of TF binding that cannot be easily detected from PWM-based motif searches or motif enrichment analysis alone. Our $F_1$-hybrid datasets identify thousands of enhancer allele pairs that differ subtly in their DNA sequences and yet have strongly allele-specific functional properties. In the future, incorporating $F_1$-hybrid data from additional cell types can further reveal both context-specific and broadly applicable mechanisms

of TF binding and enhancer activity (*Halow et al., 2021*). More broadly, this $F_1$-hybrid approach represents a powerful tool for understanding complex *cis*-regulatory processes and can accelerate efforts to identify functional non-coding variants that contribute to human disease and complex traits.

## Materials and methods

### Mice

All animal experiments were approved by the National Institutes of Health and the Harvard Medical School Institutional Animal Care and Use Committee and were conducted in compliance with the relevant ethical regulations. Six-week-old female C57BL/6 J mice were obtained from Jackson Labs (Bar Harbor, ME, USA) (Stock No. 000664) for all breeding pairs. Four- to 8-week-old male mice from the following strains were also obtained from Jackson Labs: CAST/EiJ (Stock No. 000928), MOLF/EiJ (Stock No. 000550), PWK/PhJ (Stock No. 003715), SPRET/EiJ (Stock No. 001146), 129S1/SvlmJ (Stock No. 002448), A/J (Stock No. 000646), BALB/cJ (Stock No. 000651), DBA/2 J (Stock No. 000671), NOD/ShiLtJ (Stock No. 001976). No new mouse strains were generated in this study.

The study of inbred mice that are more genetically divergent from C57BL/6 J in combination with the use of longer sequencing reads increases the proportion of informative allele-specific reads. However, higher frequencies of SNPs/indels per strain results in a greater percentage of CREs with multiple genetic variants, making it difficult to assign which specific SNP/indel is likely responsible for observed changes in TF binding or chromatin state. Therefore, to balance these considerations, we included wild-derived inbred strains with a relatively high frequency of SNPs/indels compared to C57BL/6 J mice (1 SNP/indel per ~85–170 bp), as well as more commonly used inbred strains that are less genetically divergent from C57BL/6 J mice (1 SNP/indel per ~1000 bp; *Supplementary file 1*).

### Generation of MEF lines

Embryos were harvested on embryonic day 13.5–14.5 and washed in room-temperature PBS. The heads and internal organs were removed, and the dissected tissue was re-washed in PBS. Individual embryos were placed at the center of 15 cm plates and incubated for 45 min in 1 mL trypsin-EDTA 0.25% (Life Technologies 25200072). Excess trypsin was carefully aspirated, and the dissected tissue was manually dissociated with scissors for ~1 min. Dissociated cells were then incubated in ~1 mL trypsin-EDTA at 37 °C in 5% $CO_2$ for 30 min. Complete media was prepared by supplementing DMEM (Life Technologies (Carlsbad, CA, USA) 12430062) with 10% CCS (Thermo Fisher (Waltham, MA, USA) SH3008704), Penicillin-Streptomycin (Thermo Fisher 15140148), MEM non-essential amino acids (Thermo Fisher 11140050), and 1 mM sodium pyruvate (Thermo Fisher 11360070). Trypsin was quenched with 10 mL complete media, and cells were rapidly triturated up/down 10 times with a 10 mL serological pipette to generate a single-cell suspension. An additional 10 mL complete media was added per plate, and cells were grown at 37 °C in 5% $CO_2$.

When cells became fully confluent in ~2–3 days, MEFs were washed in PBS and trypsinized in 3 mL trypsin-EDTA. A small aliquot of cells from each embryo were frozen for genotyping (see below). Cells were pelleted by spinning at 300 g and expanded onto five 15 cm plates with 20 mL complete media per plate. When fully confluent once again, MEFs were trypsinized and frozen down in freeze media (50% complete media, 40% CCS, and 10% DMSO) in aliquots of 1 plate per cryogenic vial. Cells were placed at –80 °C for ~24 hr in a cell freezing container and then transferred to liquid $N_2$ for long-term storage.

For genotyping, cells were processed with the DNeasy Blood and Tissue kit (QIAGEN (Hilden, Germany) 69506). All MEF lines were tested for mycoplasm contamination with the following primer pairs: 5'-CTTCWTCGACTTYCAGACCCAAGGCAT-3' (Myco2(cb)) with 5'-ACACCATGGGAGYTGG TAAT-3' (Myco11(cb)) and 5'-GGTGTGGGTGAGTTATTACAAARTCAATT-3' (Myco5(cb)) with 5'-GGAGTGAGTGGATCCATAAATTGTGA-3' (Myco6(cb)). Genotyping for the sex of each MEF line was performed with the following primer pair: 5'-CTGAAGCTTTTGGCTTTGAG-3' with 5'-CCACTGCC AAATTCTTTG-3'. A single 340 bp product was expected for female cells, and an additional 310 bp product was present in male cells.

## Generation of Fos antibody

We generated an in-house antibody against the full-length mouse protein for c-Fos (NCBI Reference Sequence: NP_034364.1). Briefly, we purified GST-c-Fos-His as detailed in *Sharma et al., 2019* and injected the recombinant protein into immunocompromised rabbits. Serum was collected and affinity purified using a protein A column before use in ChIP-seq and CUT&RUN experiments.

## Cell culture

Cells were thawed onto one 15 cm plate per MEF line and grown in complete media until fully confluent. For ChIP-seq and Hi-ChIP experiments, MEFs were split onto five 15 cm plates and grown in complete media until ~70–80% confluent. Cells were washed in 10 mL room-temperature PBS and switched into 20 mL warmed starve media (0.5% CCS, with the same supplement concentrations as complete media). After 26+hours in starve media, samples to be serum stimulated were incubated with 20 mL warmed stimulation media (30% CCS, with the same supplement concentrations as complete media) for 0, 10, or 90 min.

For ATAC-seq, RNA-seq, and CUT&RUN experiments, MEFs were thawed as above and were split into 6-well dishes at a concentration of $5 \times 10^5$ cells per well in 2 mL warmed starve media. Cells were grown for 26+hours, and appropriate wells were serum stimulated with 2 mL warmed stimulation media.

## Crosslinking cells

Media was aspirated from MEFs, and 2 mL or 15 mL crosslinking buffer (10 mM HEPES pH 7.5, 100 mM NaCl, 1 mM EDTA, 1 mM EGTA) with 1% formaldehyde was added for six-well or 15 cm dishes, respectively. Cells were crosslinked by shaking gently for 10 min at room temperature. Crosslinking was quenched by adding glycine to a final concentration of 125 mM and incubating for 5 min at room temperature while shaking. Cells were washed once in 2 mL or 15 mL PBS for six-well or 15 cm dishes, respectively. Cells were scraped and collected in 1 mL or 5 mL cold PBS for six-well or 15 cm dishes, respectively, and pelleted by spinning at 1000 g for 5 min at 4 °C.

## ATAC-seq libraries

MEFs from a 6-well dish were washed twice in 1 mL cold PBS and pelleted each time by spinning at 300 g for 5 min at 4 °C. A total of 50,000 MEFs were resuspended in 50 µL cold ATAC lysis buffer (10 mM Tris-HCl pH 7.5, 10 mM NaCl, 3 mM $MgCl_2$, 0.1% NP-40 0.1% Tween 20, 0.01% digitonin) and incubated for 3 min on ice. Lysed cells were washed once in 1 mL ATAC wash buffer (10 mM Tris-HCl pH 7.5, 10 mM NaCl, 3 mM $MgCl_2$, 0.1% Tween 20) by gently inverting the tube 3 times and pelleted by spinning at 500 g for 10 min at 4 °C. Pelleted nuclei were resuspended in 50 µL transposition mix (25 µL 2 x TD Buffer (Illumina (San Diego, CA, USA) 20034197), 2.5 µL TDE1 transposase (Illumina 20034197), 0.5 µL 10% Tween 20, 0.5% 1% digitonin, 16.5 µL PBS, 5 µL NF-$H_2O$) and incubated for 30 min at 37 °C with a Thermomixer set to 1,000 rpm. Samples were purified with MinElute PCR Purification Kit (QIAGEN 28004) per manufacturer's instructions and eluted in 13 µL NF-H2O. Libraries were amplified by adding the following to 10 µL purified DNA: 2.5 µL 25 µM Ad1 universal primer, 2.5 µL 25 µM Ad2.* indexing primer, 25 µL NEBNext Hi-Fi 2 x PCR Master Mix (NEB (Ipswich, MA, USA) M0541S), 10 µL NF-$H_2O$. After an initial 5 PCR cycles, libraries were quantified by qPCR by adding the following to 5 µL partially amplified DNA: 0.5 µL 25 uM Ad1 universal primer, 0.5 µL 25 µM Ad2.* indexing primer, 5 µL NEBNext Hi-Fi 2 x PCR Master Mix, 0.15 µL 1 x SYBR Green I (Thermo Fisher S7563), 3.85 µL NF-$H_2O$. All primer sequences referenced are described in *Buenrostro et al., 2015*. The number of additional PCR cycles required for amplifying remaining libraries was determined by the number of qPCR cycles needed to reach 1/3 of the maximum SYBR green signal. Libraries were purified with AMPure XP beads (0.5 x volume; Beckman Coulter (Indianapolis, IN, USA) A63881), and the supernatant was retained to remove large fragments. Primer dimers were removed by a subsequent cleanup with AMPure XP beads (1.3 x initial volume), and libraries were eluted in 20 µL NF-H2O. Libraries were sequenced on an Illumina NextSeq 500 with 40 bp paired-end reads.

## ChIP-seq libraries

Crosslinked MEFs per protocol above from 15 cm dishes were resuspended in 1 mL L1 buffer (50 mM HEPES pH 7.5, 140 mM NaCl, 1 mM EDTA, 1 mM EGTA, 0.25% Triton X-100, 0.5% NP-40, 10%

glycerol, 10 mM sodium butyrate) per 15 cm dish starting material and rotated for 10 min at 4 °C to lyse cells. Nuclei were pelleted by spinning at 1350 g for 5 min at 4 °C and resuspended in 1 mL L2 buffer (10 mM Tris-HCl pH 8.0, 200 mM NaCl, 10 mM sodium butyrate) per 15 cm dish starting material and rotated for 10 min at room temperature. Nuclei were pelleted by spinning at 1350 g for 5 min at 4 °C and resuspended in 300 µL LB3 buffer (10 mM Tris-HCl pH 8.0, 100 mM NaCl, 1 mM EDTA, 0.5 mM EGTA, 0.1% sodium deoxycholate, 0.5% N-lauroylsarcosine, 10 mM sodium butyrate) per 15 cm dish starting material. Chromatin was sonicated with a Bioruptor Plus (Diagenode (Denville, NJ, USA)) on 'high' power setting with an 'on' interval of 30 s and 'off' interval of 45 s for 36 cycles. DNA concentration was determined by taking 100 µL aliquot of sonicated chromatin, decrosslinking at 95 °C for 15 min, and purifying with QIAquick PCR Purification Kit (QIAGEN 28104) and quantifying by Nanodrop. One µg of purified chromatin in 10% glycerol was run on a 2% agarose gel and stained with ethidium bromide for 30 min to validate fragment size (typically within ~200–1,000 bp). The remainder of the sonicated chromatin was transferred to 1.5 mL tubes and centrifuged at 16,000 g for 10 min at 4 °C to pellet insoluble debris. Triton X-100 was added to soluble chromatin to a final 1% concentration. Protein A Dynabeads (Thermo Fisher 10008D) were washed twice in 1 mL cold block solution (0.5% BSA (w/v), 1% Triton X-100, diluted in LB3 buffer). For coupling antibodies to beads, 15 µL bead slurry per IP were resuspended in 1.5 mL cold block solution, and the appropriate amount of antibody (0.5 µg for anti-H3K27ac (Abcam (Waltham, MA, USA) ab4729), 0.5 µg for anti-H3K4me1 (Abcam ab8895), 0.5 µg for anti-H3K4me2 (Abcam ab7766), 0.5 µg for anti-H3K4me3 (Abcam ab8580), 2 µg for anti-Fos (in-house generated antibody and Santa Cruz Biotechnology (Dallas, TX, USA) sc-7202X), 2 µg for anti-Tead1 (Abcam ab133533), 2 µg for anti-CTCF (Active Motif (Carlsbad, CA, USA) 61312), and 2 µg for anti-JunD (Santa Cruz Biotechnology sc74)) was added before rotating beads for >2 hr at 4 °C. For pre-clearing chromatin, 15 µL bead slurry was added to appropriate amount of chromatin (40 µg for histone modifications, 80 µg for transcription factors), and additional cold LB3 buffer with 1% Triton X-100 was added such that all samples had a final volume of 1.5 mL before rotating samples for >2 hr at 4 °C. Pre-cleared chromatin was added to antibody-coupled beads, and additional cold LB3 buffer with 1% Triton X-100 was added such that all samples had a final volume of 1.8 mL before rotating samples overnight at 4 °C. 50 µL of pre-cleared chromatin was stored at –20 °C for making input libraries. For all wash steps listed below, samples were rinsed with 1 mL cold wash buffer and rotated for 5 min at 4 °C before separating beads with a magnet and discarding supernatant. Samples were washed sequentially twice in low salt buffer (0.1% SDS, 1% Triton X-100, 2 mM EDTA, 20 mM Tris-HCl pH 8.0, 150 mM NaCl), twice in high salt buffer (0.1% SDS, 1% Triton X-100, 2 mM EDTA, 20 mM Tris-HCl pH 8.0, 500 mM NaCl), twice in LiCl buffer (250 mM LiCl, 1% NP-40, 0.5% sodium deoxycholate, 1 mM EDTA, 10 mM Tris-HCl pH 8.0), and once in TE buffer (50 mM Tris-HCl pH 8.0, 10 mM EDTA). Samples were eluted from beads by addition of 200 µL TE buffer with 1% SDS and incubating at 65 °C for 30 min, with brief vortexing every 10 min to mix. IP samples were placed on magnet, and supernatant was transferred to new tubes. Input samples were also thawed, and 150 µL TE buffer with 1% SDS was added. Both IP and input samples were decrosslinked by incubating at 65 °C overnight. 10 µg RNase A (Sigma Aldrich (St. Louis, MO, USA) (R6513) was added and samples were incubated at 37 °C for 1 hr to digest RNA. 7 µL 20 µg/µL proteinase K (New England Biolabs P8107) was added, and samples were incubated at 55 °C for 2 hr to digest protein. DNA was extracted with 1 volume of 25:24:1 phenol-chloroform-isoamyl alcohol and purified with QIAquick PCR Purification Kit (QIAGEN 28104). Libraries were prepared with the Ovation Ultralow V2 DNA-Seq Library Preparation Kit (NuGEN (Redwood City, CA, USA) 0344NB-32) per manufacturer's instructions. Libraries were sequenced on an Illumina NextSeq 500 with 150 bp paired-end reads.

## CUT&RUN libraries

Crosslinked MEFs per protocol above from six-well dish were washed once in 2 mL PBS and collected in 1 mL cold NE1 buffer (20 mM HEPES pH 7.5, 10 mM KCl, 1 mM MgCl₂, 1 mM DTT, 0.1% Triton X-100, Roche Protease Inhibitor Cocktail (Millipore (Burlington, MA, USA) 11873580001)). Cells were permeabilized to isolate nuclei by rotating for 10 min at 4 °C. Nuclei were pelleted by spinning at 500 g for 5 min at 4 °C and resuspended in 1 mL cold CUT&RUN wash buffer (20 mM HEPES pH 7.5, 0.2% Tween-20, 150 mM NaCl, 0.1% BSA, 0.5 mM spermidine, 10 mM sodium butyrate, Roche Protease Inhibitor Cocktail). Ten µL concanavalin-coated bead slurry (Bangs Laboratories (Fishers, IN, USA) BP531) per sample was washed twice in 1.5 mL CUT&RUN binding buffer (20 mM HEPES pH

7.5, 10 mM KCl, 1 mM CaCl$_2$, 20 mM MnCl$_2$) and resuspended in a final volume of 10 µL CUT&RUN binding buffer per sample. After adding 10 µL bead slurry to each sample, tubes were inverted 10 times and incubated for 10 min at room temperature to bind nuclei. Beads were separated from wash buffer by placing on magnet for >30 s and were resuspended in 50 µL antibody buffer (0.1% Triton X-100, 2 mM EDTA, diluted in CUT&RUN wash buffer). Antibodies (in-house anti-Fos, anti-H3K27ac (Abcam ab4927), or rabbit IgG (Cell Signaling Technology (Danvers, MA, USA) 2729 S)) were added at 1:50 dilution, and samples were incubated overnight at 4 °C. Beads were washed once in 1 mL Triton-wash buffer (0.1% Triton X-100, diluted in CUT&RUN wash buffer) and resuspended in 50 µL antibody buffer. Protein-A MNase (*Skene and Henikoff, 2017*) was added to a final concentration of 700 ng/mL, and samples were incubated for 1 hr at 4 °C. Beads were washed twice in 1 mL Triton-wash buffer and resuspended in 100 µL Triton-wash buffer. 2 µL 100 mM CaCl$_2$ was added per sample to activate the MNase and each sample incubated on ice for 30 min. A total of 100 µL 2 x STOP buffer (340 mM NaCl, 20 mM EDTA, 4 mM EGTA, 0.1% Triton X-100, 50 µg/mL RNase A (Sigma-Aldrich R6513), 2 pg/mL yeast spike-in DNA (provided by S. Henikoff)) was added, and samples were incubated for 20 min at 37 °C to release CUT&RUN fragments from nuclei. Samples were placed on magnet, and supernatant was transferred to a new tube and added to 2 µL 10% SDS and 2 µL 20 mg/mL proteinase K (New England Biolabs P8107). Samples were incubated overnight at 65 °C to reverse crosslinks. DNA was extracted with 1 volume of 25:24:1 phenol-chloroform-isoamyl alcohol and precipitated in 2.5 volumes of 100% ethanol with 2 µL glycogen (Sigma-Aldrich 10901393001). Pellet was washed once in 1 mL 100% ethanol and dissolved in 40 µL 10 mM Tris-HCl pH 8.5. Libraries were prepared as described in *Skene and Henikoff, 2017*, with two subsequent AMPure XP bead cleanups (1.1 x volume) to fully remove contaminating adapter dimers from libraries. Libraries were sequenced on an Illumina NextSeq 500 with 40 bp paired-end reads.

## Hi-ChIP libraries

Hi-ChIP was performed as previously described in *Mumbach et al., 2017* with the following modifications. Fifteen µL of MboI restriction enzyme (New England Biolabs R0147) was used for digesting chromatin from 15 million MEFs. Sonication was performed with a Covaris M220 with the following settings: duty cycle = 5, PIP = 70, cycles/burst = 200, and time = 8 min. 75 µL of Protein A Dynabeads (Thermo Fisher 10008D) was used for IP and 1 µg of anti-H3K27ac (Abcam ab4927) antibody was used per sample to typically yield 12.5 ng DNA. Accordingly, 0.6725 µL of transposase enzyme (Nextera 20034197) was used to insert adapters, and libraries were amplified for 8 PCR cycles.

## RNA-seq libraries

MEFs from 15 cm dish were washed once in 15 mL cold PBS and pelleted by spinning at 300 g for 5 min at 4 °C. Cell pellet was resuspended in 200 µL cold cytoplasmic lysis buffer (10 mM Tris-HCl pH 7.5, 150 mM NaCl, 0.15% NP-40) and rotated for 5 min at 4 °C. Lysate was layered on top of 500 µL cold sucrose buffer (10 mM Tris-HCl pH 7.5, 150 mM NaCl, 24% sucrose (w/v)) and centrifuged at 16,000 g for 10 min at 4 °C. These steps were repeated once more to achieve higher purity in the nucleoplasmic fraction. Pelleted nuclei were resuspended in 200 µL glycerol buffer (20 mM Tris-HCl pH 7.9, 75 mM NaCl, 0.5 mM EDTA, 50% glycerol, 0.85 mM DTT), and an equal volume of cold nuclear lysis buffer was added (20 mM HEPES pH 7.5, 7.5 mM MgCl$_2$, 0.2 mM EDTA, 300 mM NaCl, 1 M urea, 1% NP-40, 1 mM DTT). Tubes were gently vortexed twice for 2 sec, incubated for 1 min on ice, and centrifuged at 18,000 g for 2 min at 4 °C. These steps were repeated once more to achieve higher purity in the chromatin fraction. The remaining chromatin pellet was resuspended in 50 µL cold PBS and vortexed briefly. 500 µL of TRIzol (Thermo Fisher 15596026) was added to the pellet and vortexed for several minutes until fully resuspended. Chromatin-associated RNA was isolated with RNeasy Mini Kit (QIAGEN 74104) per manufacturer's instructions, and libraries were generated from 250 ng starting material with NEBNext Ultra Directional RNA Library Prep Kit for Illumina (New England Biolabs E7765). Libraries were sequenced on an Illumina NextSeq 500 with 150 bp paired-end reads.

## Pseudogenome generation

SNPs occurring in the CAST/EiJ, MOLF/EiJ, PWK/PhJ, and SPRET/EiJ genomes relative to the mm10 reference genome were obtained from SNP release version 5 of the Mouse Genomes Project (*Keane et al., 2011*). Only high-confidence SNPs annotated with the PASS filter, filtered using VCFtools

(version 0.1.12; *Danecek et al., 2011*), were used in all analyses. A separate pseudogenome for each wild-derived inbred strain was constructed from these SNPs using Modtools (version 1.0.2; *Huang et al., 2014*).

## Allele-specific read mapping

Reads were trimmed with the paired-end option and with SLIDINGWINDOW:5:30 using Trimmomatic (*Bolger et al., 2014*). Paired-end reads that survived trimming were re-paired using the bbmap utility (*Bushnell, 2014*). Both unpaired and paired reads were concurrently mapped to the C57BL/6 J and appropriate pseudogenome with bowtie2 using default parameters (*Langmead and Salzberg, 2012*). Mapped reads were converted to.bam format with samtools view (*Li et al., 2009*) using the following options -h -b -F 3844 -q 10 and sorted by coordinate. Reads initially mapped to each pseudogenome were converted back to C57BL/6 J coordinates by running Lapels (*Huang et al., 2014*). All unpaired reads were then resorted by query name with samtools view -n and their flags were fixed with samtools fixmate. Informative reads (i.e. those that overlapped SNPs) were subsetted with the extractasReads.R utility from asSeq (*Sun, 2012*) and remapped to the reciprocal genome using the same commands as above. To retrieve our final set of allele-specific reads, we inputted the informative reads into the WASP pipeline (*van de Geijn et al., 2015*) to retain only those reads that map to a single locus in only one genome. Tag directories for both alleles were generated with HOMER's makeTagDirectory command with total mapped reads (i.e. before running WASP pipeline) and allele-specific reads.

For visualization purposes, mapped reads in.bam format were also converted to.bed format, and unique reads were retained (with sort -k1,1 -k2,2n -k3,3n -u) and extended by 150 bp with bedtools slop -l 0 r 150. All samples were normalized to a depth of 10 million reads, and read coverage was calculated by bedtools genomeCoverageBed. The output.bedgraph file was then converted with UCSC's bedGraphtobigWig utility, and all tracks displayed were visualized with the UCSC Genome Browser (GRCm38/mm10).

## ATAC-seq peak calling

Reads from individual bioreplicates were pooled with samtools merge. Two pseudoreplicates consisting of a random subset (50%) of total reads were generated by samtools view -h -b -s 1.5 and samtools view -h -b -s 2.5. Peaks were called from pooled reads and from two psuedoreplicates using macs2 (*Liu, 2014*) with the following options: -p 1e-1 --nomodel --extsize 200. Peaks were also called using spp (*Kharchenko et al., 2008*) with -npeak=500,000 to include a large set of putative peaks. For both macs2 and spp, reads from input DNA pooled from all ChIP-seq experiments were used as the control sample. To analyze consistency of peak calling across pseudoreplicates, we employed an Irreproducible Discovery Rate (IDR) approach (*Landt et al., 2012*) by running batch-consistency-analysis.R and ranking peaks by p.value for macs2 and signal.value for spp. Peaks with a global IDR score of 0.0025 or less were retained for downstream analyses. Since peaks called across samples from different genotypes can vary somewhat in their specific coordinates, we generated a total universe of possible ATAC-seq peaks by combining all sequencing reads into a single tag directory in HOMER (*Heinz et al., 2010*). We then used the HOMER function getPeakTags with the -center option to generate single bp coordinates with maximal ATAC-seq signal (which we refer to in our manuscript as ATAC-seq summits).

## Allele-specific CUT&RUN peak calling

Peak calling was performed as detailed above for ATAC-seq data, except reads mapping to the C57BL/6 J and corresponding pseudogenome for each $F_1$-hybrid line were inputted separately into macs2 and spp. CUT&RUN peaks were then intersected with all ATAC summits detected across all genotypes and were recentered on the summit of ATAC-seq signal. This was important to do because peak calling algorithms that we used would often identify multiple histone modification peaks for individual CREs due to the non-continuous enrichment in signal. This also enabled us to generate uniform windows centered around ATAC-seq summits to consistently quantify signal for CUT&RUN data across different ATAC-seq summits. Peaks from both the C57BL/6 J and pseudogenome were concatenated, and only peaks with at least one SNP/indel within +/-60 bp of the ATAC summit were retained for allele-specific analysis (as highly "mappable" sites). To determine whether the CUT&RUN signal is significantly skewed towards one allele, we used HOMER (*Heinz et al., 2010*) to annotate

read coverage with -noadj -size –250,250 for AP-1 and -noadj -size –500,500 for H3K27ac. These counts were inputted in DESeq2 (*Love et al., 2014*), and all peaks with an FDR <0.1 were considered allele-specific. Both allele-specific and shared peaks were then filtered by the following criteria: (1) when peak summits occurred within 1 kb of one another, only the summit with the highest pooled ATAC-seq signal was retained for downstream analyses, (2) peaks within the bottom quintile of pooled ATAC-seq signal per condition per F1-hybrid line were also excluded as low-signal sites, and (3) peaks that fell within 100 kb of gene bodies of known imprinted genes were filtered out of our remaining dataset to rule out differences in CRE activity that result from parent-of-origin effects (*Shen et al., 2014*).

## Validation of CUT&RUN experiments

Since we modified existing protocols for CUT&RUN (*Skene and Henikoff, 2017*) to decrease the number of cells and sequencing reads compared to those typically required for generating ChIP-seq data, we performed a series of analyses to ensure that we were still able to quantitatively measure TF binding. While other MNase-based methods have reported sequence-dependent biases that could result in preferential cutting at open chromatin regions (*Chung et al., 2010*), we noted a similar fraction of reads in peaks from CUT&RUN and ChIP-seq when using identical antibodies (*Figure 3—figure supplement 2D*), suggesting that we observe a minimal open chromatin bias with our modified CUT&RUN protocol. We also noted similar levels of binding at Fos peaks with ChIP-seq using our newly generated Fos antibody in comparison with a previously available commercial antibody, and we confirmed the specificity of our antibody by comparing peaks found in our Fos-binding data with HA ChIP-seq data from wild-type C57BL/6 J MEFs and C57BL/6 J MEFs that express Fos-FLAG-HA (*Figure 3—figure supplement 2A-C*). When we computationally separate shorter (<120 bp) from nucleosomal (>150 bp) Fos CUT&RUN reads, we found that sub-nucleosomal reads were more likely to be enriched at the core of Fos-bound enhancers, showed greater signal-to-noise relative to an IgG control antibody (*Figure 3—figure supplement 2F*), and could be used to detect footprints containing AP-1-binding motifs (as a proxy for the detection for sequence-specific AP-1 binding; *Figure 3—figure supplement 2G*).

## Motif footprinting with CUT&RUN reads

Since CUT&RUN utilizes an antibody-targeted MNase for cleaving DNA fragments at TF-bound regions, individual cut sites derived from both ends of paired-end sequencing reads can be used for higher resolution mapping of specific nucleotides bound by TFs within peaks. DNA motifs that are recurrently protected (termed 'footprints') from MNase by chromatin-associated protein binding were identified from Fos CUT&RUN experiments performed in serum-stimulated C57BL/6 J MEFs. Peak calling, motif identification, and footprinting analysis were performed using CUT&RUNtools (*Zhu et al., 2019*) with default parameters. Shown in *Figure 3—figure supplement 2G* is the aggregated cut site probability within +/-100 bp of all identified MTGAGTCA sites at Fos CUT&RUN peaks, suggesting that our Fos CUT&RUN experiments are able to detect direct binding sites for AP-1.

## Allele-specific ChIP-seq peak calling

Peak calling was performed as detailed above for CUT&RUN data, except we considered all ChIP-seq peaks that overlapped ATAC-seq summits. All experiments (except for CTCF ChIP-seq, which was done in cycling cells) were performed in serum-starved and restimulated MEFs (90 min) and peaks from these datasets were analyzed separately and subsequently combined for downstream analyses, with the exception of experiments that directly queried enriched motifs at stimulus-responsive enhancers (*Figure 3—figure supplement 1B*). We merged peak sets across timepoints due to the similar dynamics of enhancer activity observed across a more extensive serum stimulation timecourse in MEFs, with ~83% of enhancers exhibiting similar H3K27ac signal at 0, 10, and 90 min (*Vierbuchen et al., 2017*). Similarly, we merged our Tead1 binding data across 0 min and 90 min conditions to include as many binding sites as possible for motif analyses. This was not performed for Fos because we observed zero significant Fos peaks genome-wide in serum-starved MEFs, consistent with the fact that Fos protein is virtually undetectable in serum-starved MEFs.

### Detection of significant Hi-ChIP interactions

H3K27ac Hi-ChIP reads were aligned with HiC-Pro (*Servant et al., 2015*) using an MboI-digested mm10 genome as the reference genome. Significant H3K27ac loops were determined by running hichipper (*Lareau and Aryee, 2018*), inputting 1 kb windows centered on all previously identified distal active enhancers from C57BL/6 J MEFs (*Vierbuchen et al., 2017*) as possible loop anchors. Only loops that were supported by at least 10 paired-end tags per replicate and had a p-value <1e-4 from hichipper were retained from each timepoint (0 m and 90 m). Using these criteria, we noted similar numbers of H3K27ac Hi-ChIP loops in our dataset as those from other cell types (*Mumbach et al., 2017*). We generated tracks for visualization by retaining the midpoint of all significant loops.

### Analysis of allele-specific gene expression

Reads were mapped with STAR 2.7.3 (*Dobin et al., 2013*) with the following options to enable WASP filtering of allele-specific reads: `--outSAMattributes` NH HI AS nM vG vA `--waspOutputMode SAMtag`. Genome-specific reads were extracted and converted into.bam format with samtools view -h -b -F 3844 -q 10. The featureCounts command from Subread (*Liao et al., 2013*) was used to quantify the number of allele-specific reads per genome that overlapped each mm10 Refseq gene bodies. Genes with an average expression per sample <1 were filtered out, and counts from individual genotypes and timepoints were inputted into edgeR. Genes with an FDR <0.05 by glmQLFTest were considered allele-specific in their expression.

### Scatterplots for quantifying TF binding or chromatin state across alleles

Allele-specific reads are converted from.bam files into tag directories for HOMER (*Heinz et al., 2010*). Single bp coordinates, typically from ATAC-seq summits, are annotated with separate tag directories for the C57BL/6 J and pseudogenome-specific reads with the following options: mm10 -noadj -size −250,250 for TFs and mm10 -noadj -size −500,500 for histone modifications. The resulting read coverage values are log2 transformed and plotted with geom_point in ggplot2 against one another.

### Aggregate plots for averaging TF binding or chromatin state across peaks

Allele-specific reads were prepared as described above with HOMER. ATAC-seq peak centers or TF motif k-mers were then annotated with allele-specific read tag directories with the following options: mm10 -noadj -noann -nogene -size −1000,1000 -hist 10. Individual coverage values across 10 bp bins are plotted with geom_line in ggplot2.

### Number and position of SNPs/indels at *cis*-regulatory elements

To determine the total number of SNPs/indels within the central 150 bp of enhancers as in *Figure 2I*, we used bedtools window -w 75 c and centered on the ATAC-seq summit present at each putative CRE. We also mapped the positioning of SNPs/indels relative to the ATAC-seq summit by using bedtools window -w 200, and computed the difference in coordinates between the ATAC-seq summit and the closest nucleotide present in the SNP/indel. For defining the locations of putative CTCF motifs, we inputted the MA0139.1 profile from the JASPAR database into FIMO and limited the maximum stored scores to $10^6$ per genome. The density of SNPs/indels at regions with significant allele-specific signal was visualized (as in *Figure 2H–J* and *Figure 5A–F*) using the geom_histogram function with default parameters (from ggplot2), after centering upon the given coordinate of interest indicated on the horizontal axis. On these same plots, we plotted the smoothed density estimates for sites with allele-specific (black trace) and shared (blue trace) using the geom_density function (from ggplot2).

### Mammalian conservation scores at *cis*-regulatory elements

To more directly compare allele-specific and shared CREs with similar levels of transcriptional activity, we subsampled the pool of shared CREs such that the distribution of H3K27ac levels (+/-500 bp from ATAC-seq summit) on the active allele matched that of the allele-specific CREs. Bigwig files with phastCons scores for 60 vertebrate species for each mouse mm10 chromosome were obtained from UCSC. For each CRE, we computed a phastCons score for a 150 bp window centered on the ATAC-seq summit using the bigWigAverageOverBed script from UCSC Tools (Version 3.6.3).

## Identifying recurrent k-mer clusters at *cis*-regulatory elements with KMAC

Nucleotide sequences present at the central 60 bp of enhancers were extracted using bedtools getfasta (*Quinlan and Hall, 2010*). These.fasta files were inputted into KMAC (*Guo et al., 2018*) as the positive sequences (using the appropriate `--k_seqs [n]`) and enriched k-mer clusters are determined relative to random GC-matched control regions of equal length from the C57BL/6 J genome (using --gc –1) with the following additional options: `--k_min 5 k_max10 --k_top 10`.

## Identifying k-mers that distinguish classes of AP-1 bound sites with gkm-SVM

Coordinates for AP-1 peaks were converted to appropriate pseudogenome coordinates with modmap (*Huang et al., 2014*) with -f and -d options. Nucleotide sequences in.fasta format for both alleles of each locus (60 bp window) were obtained with bedtools getfasta and were concatenated across different $F_1$-hybrid lines. For performing the active versus inactive Fos-bound site comparison, we used the gkm-SVM package developed by Dr. Michael Beer's lab (*Ghandi et al., 2016*) and generated the kernel matrix by inputting the active allele DNA sequence (with higher H3K27ac levels) as the positive set and the corresponding inactive allele DNA sequence as the negative set. SVM training was done with the gkmsvm_trainCV command using default parameters and k-mer weights were calculated for all possible 10-mers with gkmsvm_classify.

## Acknowledgements

We thank members of the Greenberg Lab for their scientific advice and input throughout this project, L Hu for assistance in generating the anti-Fos antibody used in this study, and S Bhunia for help with data visualization. We are grateful to L Boxer, A Carter, C Davis, E Duffy, E Griffith, and E Li for their helpful feedback on this manuscript. MGY was supported by National Institutes of Health under training grants T32EY00711030 and T32AG000222. EL was supported by the National Science Foundation Graduate Research Fellowship under grant numbers DGE0946799 and DGE1144152. This work was funded by the NIH (R01 NS115965 to MEG).

## Additional information

### Funding

| Funder | Grant reference number | Author |
|---|---|---|
| National Institutes of Health | T32EY00711030 | Marty G Yang |
| National Institutes of Health | T32AG000222 | Marty G Yang |
| National Science Foundation | DGE0946799 | Emi Ling |
| National Science Foundation | DGE1144152 | Emi Ling |
| National Institutes of Health | R01 NS115965 | Michael E Greenberg |

The funders had no role in study design, data collection and interpretation, or the decision to submit the work for publication.

### Author contributions

Marty G Yang, Conceptualization, Data curation, Formal analysis, Supervision, Investigation, Methodology, Writing – original draft, Project administration, Writing – review and editing; Emi Ling, Christopher J Cowley, Conceptualization, Data curation, Formal analysis, Investigation, Methodology, Writing – original draft, Writing – review and editing; Michael E Greenberg, Conceptualization, Supervision,

Funding acquisition, Investigation, Writing – original draft, Project administration, Writing – review and editing; Thomas Vierbuchen, Conceptualization, Formal analysis, Supervision, Funding acquisition, Investigation, Methodology, Writing – original draft, Project administration, Writing – review and editing

## Author ORCIDs

Marty G Yang ⓘ http://orcid.org/0000-0001-5788-3092
Emi Ling ⓘ http://orcid.org/0000-0001-5287-0284
Michael E Greenberg ⓘ http://orcid.org/0000-0003-1380-2160
Thomas Vierbuchen ⓘ http://orcid.org/0000-0002-5690-5680

## Ethics

All animal experiments were approved by the National Institutes of Health and the Harvard Medical School Institutional Animal Care and Use Committee and were conducted in compliance with the relevant ethical regulations (Protocol # IS00000074-3).

## Decision letter and Author response

Decision letter https://doi.org/10.7554/eLife.76500.sa1
Author response https://doi.org/10.7554/eLife.76500.sa2

## Additional files

### Supplementary files

• Supplementary file 1. Total numbers of SNPs/indels per inbred mouse strain relative to the C57BL/6 J reference strain. ATAC-seq peaks were considered highly mappable if they contained a SNP/indel within a 120 bp window centered on their respective ATAC-seq summits.

• Supplementary file 2. Experimental condition, replicate number, number of sequencing reads, and percentage of non-duplicated reads for all genomic assays performed in this study.

• Supplementary file 3. List of all significant H3K27ac Hi-ChIP loops at 0 or 90 min serum stimulation in wild-type C57BL/6 J MEFs ($P$<1e-4). Only intra-chromosomal loops with at least 10 paired-end reads connecting them per bioreplicate were retained for analysis.

• Supplementary file 4. Number of allele pairs with allele-specific and shared signal for each transcription factor or histone modification surveyed in our dataset. For Fos and H3K27ac experiments, the data from ChIP-seq (wild-derived inbred strains; CAST/EiJ, MOLF/EiJ, PWK/PhJ, SPRET/EiJ) and CUT&RUN (less divergent inbred strains; 129S1/SvImJ, A/J, BALB/cJ, DBA/2 J, NOD/ShiLtJ) were merged in all rows designated "all strains". Only CUT&RUN peaks with a SNP/indel present within 60 bp of the ATAC-seq summit were included for allele-specific analyses for non-wild-derived (i.e. less genetically divergent) strains.

• Supplementary file 5. Significant allele-specific transcripts from chromatin-associated RNA-seq data using reads pooled from 0, 20, and 90 min timepoints. Positive and negative fold-changes indicate genes expressed at higher levels on the paternal, wild-derived allele and maternal, C57BL/6 J allele, respectively.

• Supplementary file 6. Location, allele-specific H3K27ac values, and DNA sequences for top decile of allele-specific enhancers, with greatest fold-change in H3K27ac signal between active and inactive alleles.

• Transparent reporting form

### Data availability

We submitted our data to GEO, and it is now accessible via GSE193728.

The following dataset was generated:

| Author(s) | Year | Dataset title | Dataset URL | Database and Identifier |
| --- | --- | --- | --- | --- |
| Yang MG, Ling E, Cowley CJ, Greenberg ME, Vierbuchen T | 2022 | Characterization of sequence determinants of enhancer function using natural genetic variation | https://www.ncbi.nlm.nih.gov/geo/query/acc.cgi?acc=GSE193728 | NCBI Gene Expression Omnibus, GSE193728 |

The following previously published datasets were used:

| Author(s) | Year | Dataset title | Dataset URL | Database and Identifier |
|---|---|---|---|---|
| Vierbuchen T, Ling E, Cowley CJ, Couch CH | 2017 | AP-1 transcription factors and the BAF complex mediate signal-dependent enhancer selection | https://www.ncbi.nlm.nih.gov/geo/query/acc.cgi?acc=GSE83295 | NCBI Gene Expression Omnibus, GSE83295 |
| Meuleman W, Stamatoyannopoulos JA | 2019 | Index and biological spectrum of accessible DNA elements in the human genome | https://doi.org/10.5281/zenodo.3838751 | Zenodo, 10.5281/zenodo.3838751 |

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
