## [Editor Report]

Here, the authors used multiple F1 crosses and the resulting embryonic fibroblasts to perform molecular profiling with ATAC-seq and a combination of ChIP-seq, Hi-ChIP, and CUT&RUN on multiple modified histones and transcription factors proteins. These important results are a convincing resource for quantifying allelic bias in protein-DNA binding and chromatin accessibility.

---

## [Decision Letter]

**Decision letter after peer review:**

Thank you for submitting your article "Characterization of sequence determinants of enhancer function using natural genetic variation" for consideration by *eLife*. Your article has been reviewed by 2 peer reviewers, one of whom is a member of our Board of Reviewing Editors, and the evaluation has been overseen by Detlef Weigel as the Senior Editor. The reviewers have opted to remain anonymous.

Essential revisions:

We all thought the data are quite strong. We have included the full review from both reviewers below and have summarized the essential three items for revision here:

1) The authors claim "Our findings provide new insight into how enhancer function is encoded within DNA sequences…" but it was hard to see the new insights that were provided. We suggest hardening up the abstract to clarify precisely what the authors have discovered, and considering reorganizing the text to consolidate and focus the introductory material on what is relevant to these key claims.

2) The overall data set is strong, but they are not utilized to the fullest extent for analyses. For example, no population genetics data were considered. Can the authors examine allele frequency and effect size relationships? Basic population genetics tells us that rare SNPs are more likely to be deleterious. So, in situations where C57BL/6J has the rare allele -- is accessibility and TF binding more likely to be reduced? There are several analyses that could address this general question.

3) Data sharing through GEO must be complete.

*Reviewer #1 (Recommendations for the authors):*

The authors claim "Our findings provide new insight into how enhancer function is encoded within DNA sequences…" but I had trouble seeing the new insights that were provided here.

Several claims were made about enhancer priming, selection, and other chromatin kinetics. But to really understand this, I think one needs time series data, which is not part of the present study.

In the Allele-specific CUT&RUN peak calling section, what does it mean to recenter a CUT&RUN peak onto an ATAC-seq summit? Are you shifting the coordinates of the CUT&RUN peak so the middle occurs where the ATAC summit is located? Does this make sense to do? Given that a modified histone should not occur at the same place as an open chromatin summit, I'm not sure why this was done. Perhaps there are missing details that should be here.

It looks like there are issues with the GEO upload and these should be resolved before the manuscript is finalized.

The abstract claim of "impact of sequence variation on enhancer function" seems a little broad.

Sometimes, the figures should be labeled more clearly. For example, Figure 6A-B has two different line colors but these are not explained with a color key in the figure.

Figure 2H-J and every panel in Figure 5 have typos on the x-axis. -100 is incorrectly plotted as -10.

The acronym CRE is used extensively throughout the paper, beginning in the second sentence of the introduction, but is never defined.

*Reviewer #2 (Recommendations for the authors):*

1) The abstract is pretty general and did not help me figure out what the main claims of the paper were. I had difficulty triangulating the claim I found most interesting with its description in the narrative and the data in the figures. For example, the abstract claims "our data…reveal a hierarchical relationship between AP-1 and TEAD TF binding at enhancers" (lines 38-39), which is too vague. The relevant section begins on page 21, but includes two paragraphs of introduction before arriving at the new claim. I would suggest hardening up the abstract to clarify precisely what the authors have discovered, and considering reorganizing the text to consolidate and focus the introductory material on what is relevant to these key claims.

2) I found the use of the term "k-mer" confusing. It appears that the authors use this as shorthand for "TF recognition sequence" or "motif match", but sometimes as meaning TF consensus/motif (e.g. the AP1 consensus in line 666). The gk-SVM uses another definition, where k-mers are used as SVM features. I eventually gathered that the first use follows from the KMAC (Guo et al. 2018) tool. I would suggest clarifying this terminology to refer more generally to TF recognition sequences where appropriate.

3) The manuscript would benefit from a supplemental table describing the details of the sequencing data generated, including basic experimental metadata, sequencing QC statistics (read counts, duplicate rates, enrichment, etc.), and identification of replicates.

4) The authors say that the data have been submitted to GEO but the accession was not available upon submission. The GEO record would need to be verified as providing the sequencing data underlying the manuscript in a well-organized fashion.

5) Several figures include data and trend lines which are not clearly described in the legends (e.g. the lines vs. the bars in Figure 2H-J, Figure 5, and Figure 6).

6) The authors mention that experiments were performed in both serum-starved and stimulated cells. There are occasional references to differences between these conditions (e.g. line 393, line 702), but it appears the authors generally decided to lump data from both conditions together. The authors should provide an assessment of the similarity of the two conditions, and clearly describe and justify in the narrative how they handled this variable.

---

## [Author Response]

Essential revisions:We all thought the data are quite strong. We have included the full review from both reviewers below and have summarized the essential three items for revision here:1) The authors claim "Our findings provide new insight into how enhancer function is encoded within DNA sequences…" but it was hard to see the new insights that were provided. We suggest hardening up the abstract to clarify precisely what the authors have discovered, and considering reorganizing the text to consolidate and focus the introductory material on what is relevant to these key claims.

Based on the feedback from the reviewers, we have modified the abstract to better highlight the new findings in the manuscript. We have also extensively edited the introduction of the revised manuscript so that it is more focused.

2) The overall data set is strong, but they are not utilized to the fullest extent for analyses. For example, no population genetics data were considered. Can the authors examine allele frequency and effect size relationships? Basic population genetics tells us that rare SNPs are more likely to be deleterious. So, in situations where C57BL/6J has the rare allele -- is accessibility and TF binding more likely to be reduced? There are several analyses that could address this general question.

It would be interesting to examine the relationship between allele frequency and effect size relationships, however, we do not think that our experimental design is appropriate for this analysis because all of our F_1_-hybrid cells were generated by crossing inbred mouse lines to C57BL/6J mice. Each inbred strain was originally derived from a relatively small number of mice from wild populations that were subsequently inbred for many generations. Thus, each inbred strain is homozygous for most alleles across the genome, and as a result is only representative of a small fraction of the diversity of alleles present in the natural population from which these mice were first caught. The allele frequency across each of these inbred strains is thus not indicative of their frequency in natural populations. In the future, it would be interesting to examine this question in a more appropriate context, such as commercially available outbred mouse strains. This is mentioned in the revised discussion (Lines 740-745).

3) Data sharing through GEO must be complete.

We thank the reviewers for reminding us about this critical issue. All of the data is now uploaded and publicly available via the following accession number: GSE193728. The GEO accession number has also been added to the revised manuscript (Lines 1313-1314).

Reviewer #1 (Recommendations for the authors):The authors claim "Our findings provide new insight into how enhancer function is encoded within DNA sequences…" but I had trouble seeing the new insights that were provided here.Several claims were made about enhancer priming, selection, and other chromatin kinetics. But to really understand this, I think one needs time series data, which is not part of the present study.

We have removed the discussion of chromatin kinetics at enhancers from the revised manuscript.

In the Allele-specific CUT&RUN peak calling section, what does it mean to recenter a CUT&RUN peak onto an ATAC-seq summit? Are you shifting the coordinates of the CUT&RUN peak so the middle occurs where the ATAC summit is located? Does this make sense to do? Given that a modified histone should not occur at the same place as an open chromatin summit, I'm not sure why this was done. Perhaps there are missing details that should be here.

We apologize for not explaining this in sufficient detail. In the Methods section of the revised manuscript, we have included a more detailed description of how we uniformly called ATAC-seq peaks across different genotypes and our rationale for doing so, reproduced here (Lines 1126-1131):

“Since peaks called across samples from different genotypes can vary somewhat in their specific coordinates, we generated a total universe of possible ATAC-seq peaks by combining all sequencing reads into a single tag directory in HOMER (Heinz *et al*., 2010). We then used the HOMER function getPeakTags with the -center option to generate single bp coordinates with maximal ATAC-seq signal (which we refer to in our manuscript as ATAC-seq summits).”

Moreover, we clarified what we intended by recentering a CUT&RUN peak onto an ATAC-seq summit, reproduced here (Lines 1134-1140):

“Peak calling was performed as detailed above for ATAC-seq data, except reads mapping to the C57BL/6J and corresponding pseudogenome for each F_1_-hybrid line were inputted separately into macs2 and spp. CUT&RUN peaks were then intersected with all ATAC summits detected across all genotypes and were recentered on the summit of ATAC-seq signal. This was important to do because peak calling algorithms that we used would often identify multiple histone modification peaks for individual CREs due to the non-continuous enrichment in signal. This also enabled us to generate uniform windows centered around ATAC-seq summits to consistently quantify signal for CUT&RUN data across different ATAC-seq summits.”

It looks like there are issues with the GEO upload and these should be resolved before the manuscript is finalized.

Since the initial submission, we have uploaded all of our raw and processed data, and these data are now publicly available via the following accession number: GSE193728. This accession number has been added to the revised manuscript as well (Lines 1313-1314).

The abstract claim of "impact of sequence variation on enhancer function" seems a little broad.

We have replaced that sentence with a more specific statement:

“Taken together, these data represent one of the most comprehensive assessments of allele-specific TF binding and enhancer function to date and reveal how sequence changes at enhancers alter their function across evolutionary timescales.”

Sometimes, the figures should be labeled more clearly. For example, Figure 6A-B has two different line colors but these are not explained with a color key in the figure.

We have modified the figures to include further description of these important details, as suggested. We added a color key in Figure 6A-B now to indicate that the black and green traces represent the area under curve values for Fos peaks in the C57BL/6J genome with and without masking core AP-1 sites, respectively.

Figure 2H-J and every panel in Figure 5 have typos on the x-axis. -100 is incorrectly plotted as -10.

We have corrected this mistake in the revised manuscript.

The acronym CRE is used extensively throughout the paper, beginning in the second sentence of the introduction, but is never defined.

In the revised manuscript, we now define cis-regulatory elements (CREs) in Lines 49-54 and 176-178 of the manuscript.

Reviewer #2 (Recommendations for the authors):1) The abstract is pretty general and did not help me figure out what the main claims of the paper were. I had difficulty triangulating the claim I found most interesting with its description in the narrative and the data in the figures. For example, the abstract claims "our data…reveal a hierarchical relationship between AP-1 and TEAD TF binding at enhancers" (lines 38-39), which is too vague. The relevant section begins on page 21, but includes two paragraphs of introduction before arriving at the new claim. I would suggest hardening up the abstract to clarify precisely what the authors have discovered, and considering reorganizing the text to consolidate and focus the introductory material on what is relevant to these key claims.

See response to Essential revisions.

2) I found the use of the term "k-mer" confusing. It appears that the authors use this as shorthand for "TF recognition sequence" or "motif match", but sometimes as meaning TF consensus/motif (e.g. the AP1 consensus in line 666). The gk-SVM uses another definition, where k-mers are used as SVM features. I eventually gathered that the first use follows from the KMAC (Guo et al. 2018) tool. I would suggest clarifying this terminology to refer more generally to TF recognition sequences where appropriate.

We have clarified our definition of k-mers in the revised manuscript (Lines 366-370). When referring to specific TF binding motifs, we have replaced k-mer with “motif”.

The output of the KMAC algorithm is also “k-mer based motif representation”, which are nucleotide sequences of variable length (k) that are enriched within CREs. k-mer based motif representation is an alternative method for representing TF binding motifs, which preserves positional dependencies between nucleotides (as described in Guo *et al.*, 2018). This is distinct from the more commonly used position weight matrices (PWM) representation, because k-mers explicitly account for the occurrence of particular nucleotide identities at each position, whereas PWM representations of motifs indicate the probability that a given nucleotide will be observed at a given position. This is indicated now in the text in Lines 378-383.

The output from the gkm-SVM algorithm that we employed also includes a set of kmers, and we have clarified that the use of k-mer in this context is not necessarily referring to a TF-binding motif, but instead a sequence of length (k) identified as predictive by the gkmSVM model. We have modified the text in this section to clarify this point (Lines 618-620).

3) The manuscript would benefit from a supplemental table describing the details of the sequencing data generated, including basic experimental metadata, sequencing QC statistics (read counts, duplicate rates, enrichment, etc.), and identification of replicates.

In the revised manuscript, we now include a supplementary table (Table 6) that includes the details of all of the sequencing data generated for this study (cited on Line 370 of the revised manuscript).

4) The authors say that the data have been submitted to GEO but the accession was not available upon submission. The GEO record would need to be verified as providing the sequencing data underlying the manuscript in a well-organized fashion.

Since the initial submission, we have uploaded all of our raw and processed data, and this should be viewable with the following accession number: GSE193728. This accession number has been added to the revised manuscript as well (Lines 13131314).

5) Several figures include data and trend lines which are not clearly described in the legends (e.g. the lines vs. the bars in Figure 2H-J, Figure 5, and Figure 6).

We have modified the figure legends to include complete description of these features (Lines 1253-1258 and 1445-1447).

6) The authors mention that experiments were performed in both serum-starved and stimulated cells. There are occasional references to differences between these conditions (e.g. line 393, line 702), but it appears the authors generally decided to lump data from both conditions together. The authors should provide an assessment of the similarity of the two conditions, and clearly describe and justify in the narrative how they handled this variable.

We have added a subsection in our Methods (“Allele-specific ChIP-seq peak calling”) to detail how we handled data from different timepoints and our reasoning for when we would compile data between serum-starved and stimulated MEFs (Lines 1194-1206).

In general, all experiments were performed in both serum-starved (0 min) and restimulated MEFs (90 min), with the exception of our CTCF binding dataset that was generated in asynchronous, proliferating MEFs. For all analyses, we defined active enhancers and promoters based on the presence of an ATAC-seq peak with H3K27ac enrichment at either the 0 min or 90 min timepoint. This decision was informed by data from a previous publication (Vierbuchen *et al.*, 2017), in which we noted that the vast majority of active MEF enhancers (~83%) exhibited similar levels of H3K27ac enrichment at 0, 10, or 90 min of serum stimulation.

In this manuscript, we focused on identifying sequence properties of active enhancers identified in both conditions (0 or 90 min serum stimulation). The only analysis where we separately considered the set of enhancers that inducibly gained H3K27ac upon 90 min of serum stimulation was the KMAC search we performed to identify enriched binding motifs at late-response gene enhancers (Figure 3 —figure supplement 1B).

Moreover, for the TF-binding data, we only considered FOS peaks present after 90 min of serum stimulation because Fos expression is induced by serum stimulation, and Fos protein is undetectable in serum-starved MEFs (0m). On the other hand, TEAD1 peaks in our dataset were defined based on an enrichment of signal at either 0 or 90 min, since it expressed robustly at both timepoints.